# Periodic spinodal decomposition in double–strengthened medium–entropy alloy

Hyojin Park [1,2], Farahnaz Haftlang [2,3,4] ✉, Yoon–Uk Heo [3] ✉, Jae Bok Seol [5], Zhijun Wang [6] & Hyoung Seop Kim [2,3,7,8] ✉

Achieving an optimal balance between strength and ductility in advanced engineering materials has long been a challenge for researchers. In the field of material strengthening, most approaches that prevent or impede the motion of dislocations involve ductility reduction. In the present study, we propose a strengthening approach based on spinodal decomposition in which Cu and Al are introduced into a ferrous medium–entropy alloy. The matrix undergoes nanoscale periodic spinodal decomposition via a simple one-step aging procedure. Chemical fluctuations within periodic spinodal decomposed structures induce spinodal hardening, leading to a doubled strengthening effect that surpasses the conventional precipitation strengthening mechanism. Notably, the periodic spinodal decomposed structures effectively overcome strain localization issues, preserving elongation and doubling their mechanical strength. Spinodal decomposition offers high versatility because it can be implemented with minimal elemental addition, making it a promising candidate for enhancing the mechanical properties of various alloy systems.

Owing to their unique composition, comprising multiple high-concentration elements, high/medium entropy alloys (H/MEAs) offer a plethora of alloy design opportunities as a novel class of advanced metallic materials[1–8]. H/MEAs induce further enhanced strengthening and mechanical properties through solid solution strengthening[3]. Specifically, the precipitation strengthening mechanism provides a fundamental method for enhancing strength by directly obstructing dislocation motion. This strengthening process involves the management of various factors, including the addition of minor elements to promote precipitation, aging conditions, and the interface relationship between the matrix and precipitates. Recent studies have demonstrated strength advancements by carefully manipulating the precipitation process[2,7,9,10]. However, despite this delicate control, the formation of precipitates typically results in strain localization, causing a ductility reduction of ~10–20% owing to local plastic instability[11,12]. This trade-off between strength and ductility highlights the requirement for alternative approaches to enhance the mechanical properties of face-centered cubic (FCC)-based H/MEAs while preserving their ductility.

Spinodal decomposition is a spontaneous process in which phase separation occurs without an energy barrier for nucleation with a spontaneous growth of amplitude fluctuation in composition[13]. The exclusive features of spinodal decomposition lie in its ability to create finely distributed coherent structures within the matrix. Nanoscale

[1]Department of Materials Science and Engineering, Pohang University of Science and Technology, Pohang, Republic of Korea. [2]Center for Heterogenic Metal Additive Manufacturing, Pohang University of Science and Technology, Pohang, Republic of Korea. [3]Graduate Institute of Ferrous & Eco Materials Technology, Pohang University of Science and Technology, Pohang, Republic of Korea. [4]Department of Materials Science & Engineering, Northwestern University, Evanston, IL, USA. [5]Department of Materials Engineering and Convergence Technology, Center for K–Metal & Microscopy, Gyeongsang National University, Jinju, South Korea. [6]State Key Laboratory of Solidification Processing, Northwestern Polytechnical University, Xi'an, China. [7]Advanced Institute for Materials Research (WPI–AIMR), Tohoku University, Sendai, Japan. [8]Institute for Convergence Research and Education in Advanced Technology, Yonsei University, Seoul, Republic of Korea. ✉e-mail: farahnaz.haftlang@gmail.com; yunuk01@postech.ac.kr; hskim@postech.ac.kr

coherent boundaries have been proposed to enhance strength and ductility[6,14]. The contribution of spinodal decomposition has also been investigated to improve the strength and hardness of predominantly binary and ternary systems[15–17]. However, unknown spinodal decomposition characteristics in multi-component alloy systems limit its application in the current research area. Therefore, bridging the knowledge gap between spinodal hardening and multi-component alloys can provide further insights into the fundamental principles of spinodal hardening with clear potential to further enhance the mechanical properties of advanced alloys.

In this work, to realize spinodal decomposition in multi-principal element alloys, we designed a ferrous medium entropy alloy ($Fe_{61.75}Ni_{14.25}Co_{7.6}Mn_{7.6}Ti_{2.85}Si_{0.95}Cu_{4.5}Al_{0.5}$ denoted as Fe–MEA) by adding the minor elements Cu and Al. This deliberate compositional modification introduces the formation of a miscibility gap and subsequent spinodal decomposition. This complex compositional strategy not only increases the entropy of the alloy but also enhances its enthalpy, thereby improving the probability of spinodal decomposition[18]. Cu, which has a large miscible gap with Fe—the principal component of the alloy—was added 4.5%. Adding over 5% of Cu is avoided due to the formation of microscale phase separation and immoderate Cu-rich precipitates[19]. Al was deliberately selected at a concentration of 0.5 at% to strategically leverage the benefits of lattice expansion and solid solution strengthening. Both Cu and Al are widely known to increase the stacking fault energy, preventing the generation of transformation-induced plasticity[20]. Based on the phase diagram (Supplementary Fig. 1), the presence of $Fe_2SiTi$, $Ni_3Ti$, B2, and Cu-rich FCC phases is possible upon suitable heat treatment in the range of ~450–600 °C; thus the appropriate heat treatment should be performed to achieve the desired spinodal decomposition and the formation of multiple dynamic precipitates simultaneously (detail describes in Supplementary Note 1 and "Methods" section). Consequently, Fe–MEA utilizing spinodal decomposition shows a doubled strength without notable ductility loss. Leveraging spinodal decomposition holds significant promise for advancing alloy design strategies and overcoming the long-standing challenge of balancing strength and ductility in H/MEAs, opening up different horizons for developing next-generation metallic materials.

## Results and discussion

### Initial microstructure of the aged sample

Figure 1a, b shows the electron backscatter diffraction (EBSD) phase and inverse pole figure (IPF) maps of the annealed sample. The phase of this sample was nearly fully FCC, with a small amount of martensite thermally induced by water quenching. Additionally, it exhibited a partially recrystallized microstructure with fine grains and an average size of ~4.35 μm. The volume fraction of the recrystallized grains was ~30%, with predominant nucleation along the shear bands[21] and grain boundaries[22]. The magnified images of the annealed sample in Fig. 1b1, b2 reveal a discernible difference in kernel average misorientation (KAM) value between the recrystallized and non-recrystallized regions. The microstructure of the aged samples is shown in Supplementary Fig. 2. Notably, it reveals that the annealed and the aged samples have similar microstructures on the microscale because the aging temperature of 550 °C is not high enough to induce recrystallization.

Further microstructural analysis via transmission electron microscopy (TEM) revealed nanoscale microstructural evolution during aging. As shown in the TEM bright-field image in Fig. 1c, the aged sample inherited the original microstructure-annealed state, containing both recrystallized and non-recrystallized regions. Scanning transmission electron microscopy (STEM) images and the corresponding energy–dispersive X-ray spectroscopy (EDS) maps (Fig. 1c1, c2) revealed a periodic spinodal decomposition structure with Fe, Cu, Ni, and Ti distributed throughout the matrix. The chemical

composition separation between Fe and Cu is apparent due to a practical limit of ~0.21 wt.% at 550 °C[23].

The TEM/STEM and atom probe tomography (APT) images revealed several distinct compositional heterogeneities in the formation of precipitates with distinctive elemental distributions. This includes the discovery of $Fe_2SiTi$ and $Ni_3Ti$ nanoprecipitates, which are abundant precipitates in the alloy system, alongside the observation of $Ni_3(Ti, Si)_2$, a metastable phase prone to transformation under prolonged aging[24]. Additionally, elemental segregation near dislocation lines and at grain boundaries was identified, highlighting the diffusion paths that favor precipitation growth. This segregation leads to the formation of distinct phases and clusters, such as Fe-rich BCC, Cu-rich FCC precipitates, and Cu clusters—are explained in detail in Supplementary Figs. 3–7 and Supplementary Note 2.

Spinodal decomposition is a spontaneous phenomenon in alloys with a miscibility gap that reduces the free energy of the system by forming a periodic structure characterized by small composition fluctuations within the matrix. In contrast to the nucleation and growth processes, which involve large compositional fluctuations within the microstructure and have a preferred location site such as grain boundary and dislocation due to the low energy barrier for formation, spinodal decomposition exhibits subtle compositional variations across the entire matrix[13] due to the absence of energy barrier.

An APT investigation was conducted to obtain atomic insights into the spinodal decomposition of the aged sample. As a result of spinodal decomposition, compositional modulation with lattice misfit[25], particularly in the Cu-rich modulated structure—is observed in Figs. 2 and 3. The modulated structure exhibited sideband peaks in the synchrotron X-ray diffraction (XRD) patterns (Supplementary Fig. 8). The sidebands had an asymmetric nature, and the observed shift of the sideband peaks toward the main Bragg peak indicated an increase in the wavelength associated with compositional modulation[26,27].

Based on the reconstructed APT interpretation, the isosurface maps for the two elements are shown in Fig. 2a–c, with Ti and Ni strongly tangled with each other. Upon closer inspection, a core-shell structure was observed, with Ni covering Ti, as shown in Fig. 2a2. The formation of the core-shell structure is influenced by the mixing enthalpy of the constituent elements. During aging, the negative enthalpy of mixing between Ni and Ti drives the formation of a Ni–Ti-rich region. The atomic fraction of Ni is more than 4 times higher than that of Ti in this alloy. Most of the Ti combines with Ni to form the Ni–Ti-rich region, occupying all available vacancies. Consequently, the excess Ni, which tends to interact with Ti to form a uniform structure, aggregates around the Ni–Ti bonding to constitute the shell. The core-shell structure likely resulted from uphill diffusion, thus providing indirect evidence for spinodal decomposition[28]. Though, the formation kinetics of the Ni–Ti core-shell nanoprecipitates in detail are beyond the scope of the current research. As shown in Fig. 2b, c, Ni, and Ti were located near Cu but did not interact with it. A reconstructed three-dimensional (3D) APT video of Cu, Ni, and Ti is provided in Supplementary Video 1. As Cu had a positive mixing enthalpy with almost all the constituent elements in the alloy, it exhibited a more spherical shape than the other elements, reducing the total interfacial energy[29].

High-resolution TEM (HRTEM) was employed to investigate the matrix morphology (Fig. 2d) and observe the B2 precipitates in the matrix. Two directional moiré fringes were observed at ~1 nm owing to the overlapping of the matrix and B2 phase lattice fringes. The corresponding fast Fourier transform (FFT) pattern (Fig. 2d1) exhibited discernible characteristic spots originating from the moiré fringes of the embedded B2 particles having two variants of the Kurdjumov–Sachs orientation relationship with the FCC matrix. A detailed schematic and description of the two variants of the Moiré patterns in the FFT pattern are presented in Supplementary Fig. 9, which shows the sequential steps. The specimen thickness affects the

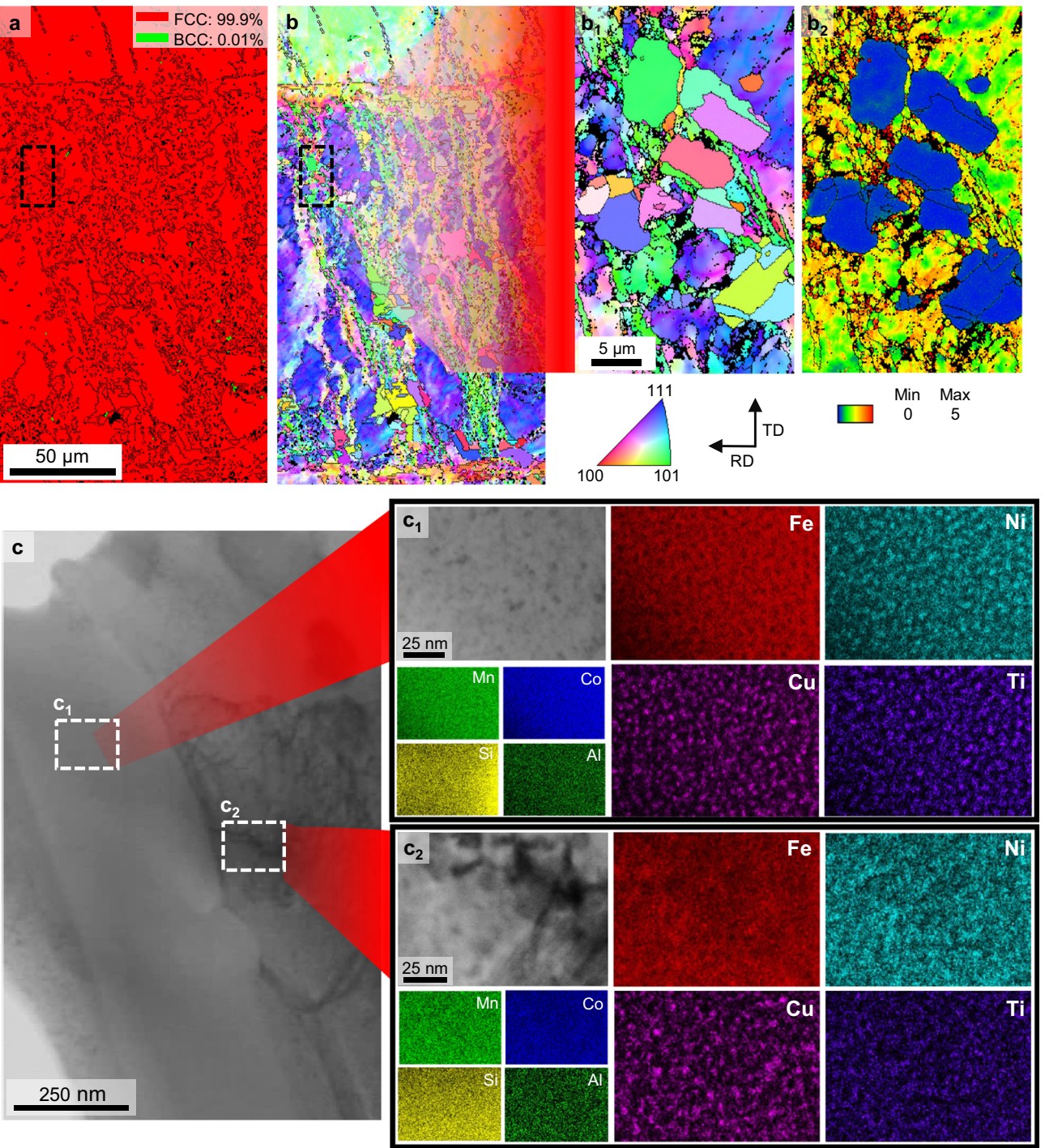

**Fig. 1 | Compositional heterogeneities of the Fe−MEA.** EBSD (**a**) phase and (**b**) IPF maps of the aged samples. Magnified (**b₁**) IPF and (**b₂**) KAM maps of an aged sample. **c** TEM bright-field image of the initial microstructure of the aged sample, along with STEM images and corresponding EDS maps of the (**c₁**) recrystallized region and (**c₂**) non-recrystallized region magnified in (**c**).

Moiré pattern generation, as shown in Supplementary Fig. 10. Two specimen thicknesses (44 nm and 54 nm) were selected for the comparison. The specimen thickness was measured using the low-loss spectra in electron energy loss spectroscopy (EELS). The detailed method of measuring specimen thickness is given in the previous report[30]. The STEM bright-field image in Supplementary Fig. 10b, where the thickness is about 44 nm, shows isolated and overlapped distributions of the nanoprecipitates. This tendency coincides with the EDS Ti map in Supplementary Fig. 10b₁. The HRTEM image was obtained in the same area (Supplementary Fig. 10b₂). The Moiré

patterns caused by the overlap of B2 and matrix reflect the projected distribution of B2 particles in the matrix: some particles are isolated, and others are overlapped. However, the nanoprecipitates mostly overlap in the thicker region (54 nm) (Supplementary Fig. 10c, c₁). The Moiré patterns in this area reveal the overlapping of B2 particles in Supplementary Fig. 10c.

To quantify the fluctuation in the chemical composition, a composition profile was obtained along the elastically soft $[001]_{FCC}$ direction, which is the strongly preferred direction for spinodal decomposition[31]. Fig. 3 presents the result of the APT reconstruction

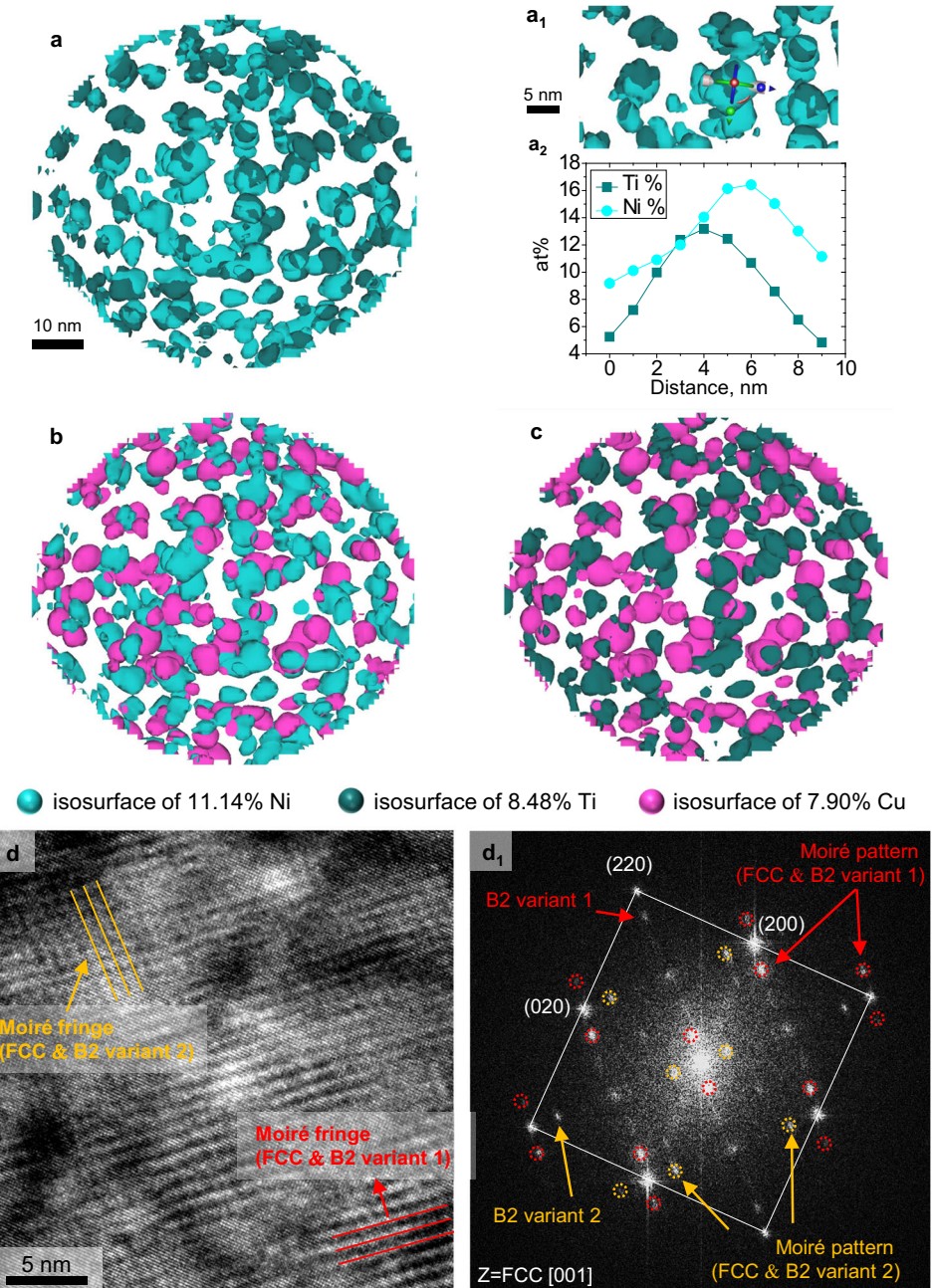

● isosurface of 11.14% Ni    ● isosurface of 8.48% Ti    ● isosurface of 7.90% Cu

**Fig. 2 | Periodic spinodal decomposed structure.** Spinodal decomposed structure: top-down view of reconstructed APT images of (**a**) Ni–Ti, (**b**) Ni–Cu, and (**c**) Cu–Ti; (**a₁**) magnified images of (**a**) ; **a₂** the Ni–Ti composition distribution profile from a cylinder in (**a₁**). **d** HRTEM image of the matrix and (**d₁**) corresponding FFT pattern with the FCC matrix along the [001]) zone axis with indexing moiré patterns originating from overlapping of two B2 having variants of the Kurdjumov–Sachs orientation relationship with the matrix (as explained in Supplementary Fig. 9).

from the aged sample. Figure 3a shows the reconstructed atom maps for Ni, Ti, and Cu, which show the modulated structure to be consistent with the previous results of Figs. 1 and 2. Figure 3b is a hit detector map obtained from the region marked in Fig. 3a, which reveals the crystallographic poles. The one-dimensional chemical composition profile of Fig. 3c is obtained along the [001] direction. The Fe, Cu, Ti, and Ni compositional fluctuations exhibited a sinusoidal pattern, with an average amplitude range of 0.32–1.30%. The wavelength of these fluctuations was 11 nm, according to the analysis shown in Fig. 3c. Notably, Ni–Ti exhibited a similar tendency to segregate from Fe to Cu. However, the precise segregation pattern of Ni–Ti differed from that of Cu, suggesting an individual trend. Apart from the Fe– and Cu–rich regions, Ni and Ti were segregated because of their strong interaction

resulting from the negative mixing enthalpy (Supplementary Table 1). The frequency distribution analysis results obtained from APT data, as presented in Supplementary Table 2, reveal a strong non–random distribution i.e., a sinusoidal pattern of Fe, Ni, Ti, Si, and Cu. It is speculated that the strong interaction among Ni, Ti, and Si, leading to the formation of the B2 structure[32,33], as observed in Supplementary Fig. 7b, c. According to the HRTEM and APT results, along with Supplementary Table 2, spinodal decomposition occurred owing to the separation of Cu and Fe within the FCC structure. Additionally, the Ni(Ti, Si) B2 phase is the result of spinodal decomposition.

During the aging process, as the multi-principal element alloy moves toward an equilibrium state, it develops segregation at the grain boundaries, various precipitates, and a periodic spinodal decomposed

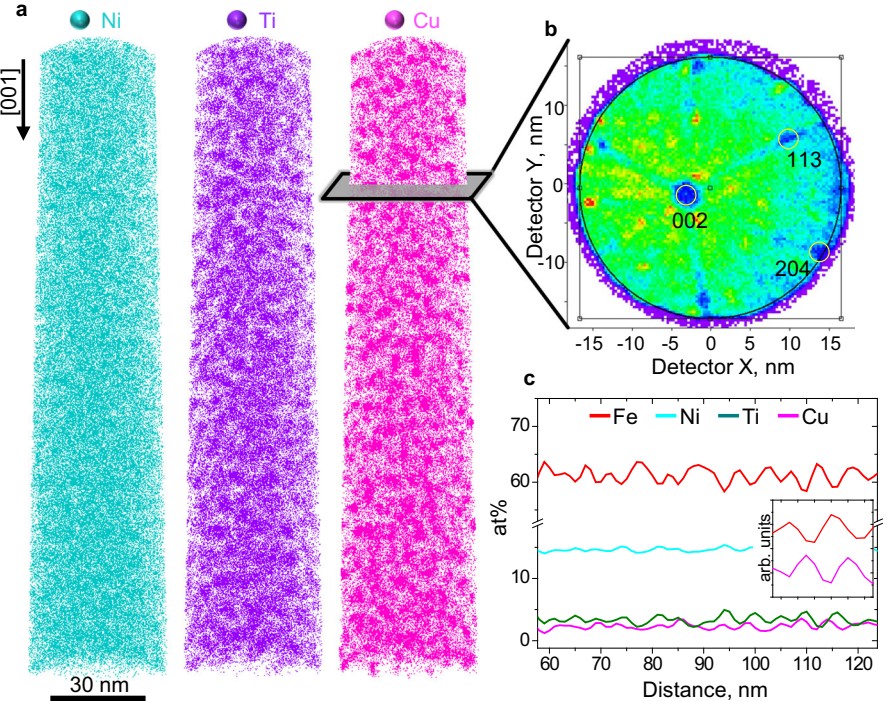

**Fig. 3 | APT reconstruction of the aged sample. a** APT reconstructed maps of alloying species in the studied alloy, showing the modulated structure. **b** Hit detector map of the region marked by a black box, oriented along the [001] direction. Crystallographic main poles are indexed on the map. **c** One-dimensional concentration profile along the $[001]_{FCC}$ and normalized composition profiles for Fe and Cu, with the maximum value of each element set to 1 for comparison in (**c**).

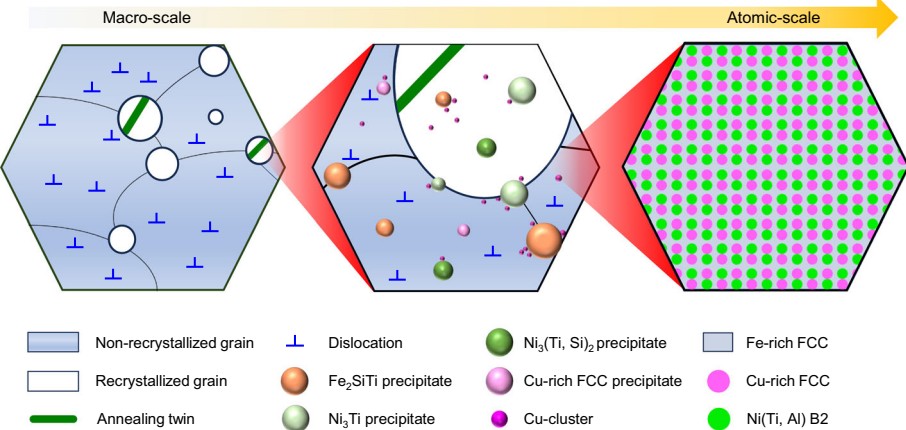

**Fig. 4 | Schematic of the aged sample.** Schematic depicting the intricate heterogeneity manifested across multiple scales in the system under investigation.

structure. Figure 4 shows a schematic image of the aged sample with complex heterostructures from micro to atomic scales obtained with simple thermo-mechanical processing.

## Mechanical properties

Figure 5a, b present representative tensile stress-strain and strain hardening rates versus true strain curves for the annealed and aged samples. The annealed sample had a yield strength, ultimate tensile strength, and total elongation of 583 MPa, 778 MPa, and 31.4%, respectively. The yield strength of the aged sample was nearly doubled, i.e., up to ~1090 MPa, with a favorable elongation of 28.5%. The ultimate tensile strength of the aged sample was ~1310 MPa, indicating that the strength was doubled without compromising ductility. Such an extraordinary combination of high mechanical strength and ductility has not been previously reported for FCC–based ferrous H/MEAs; see Fig. 5c. Therefore, microstructural architecting through

microstructural heterogeneity and multiple dynamic precipitations upon aging led to a distinguished improvement in strength with no significant change in ductility through a simple aging process. Figure 5c presents impressive accomplishments in H/MEAs and conventional alloys reported in the recent literature.

## Strengthening mechanism

A TEM analysis was conducted on a post-deformation sample at a local true strain of 5% to investigate the combination of high strength and ductility in the aged sample. Figure 5d–$d_3$ show bright and dark field images and the corresponding selected-area electron diffraction (SAED) patterns of the deformed aged sample. Profuse deformation nano-twins with an average thickness of 17.39 ± 10.98 nm (calculated from Fig. 5d), which contributed to the increased strength through dynamic Hall–Petch strengthening[34,35], were detected in the early deformation stage. EBSD investigation with twin boundary indexing on

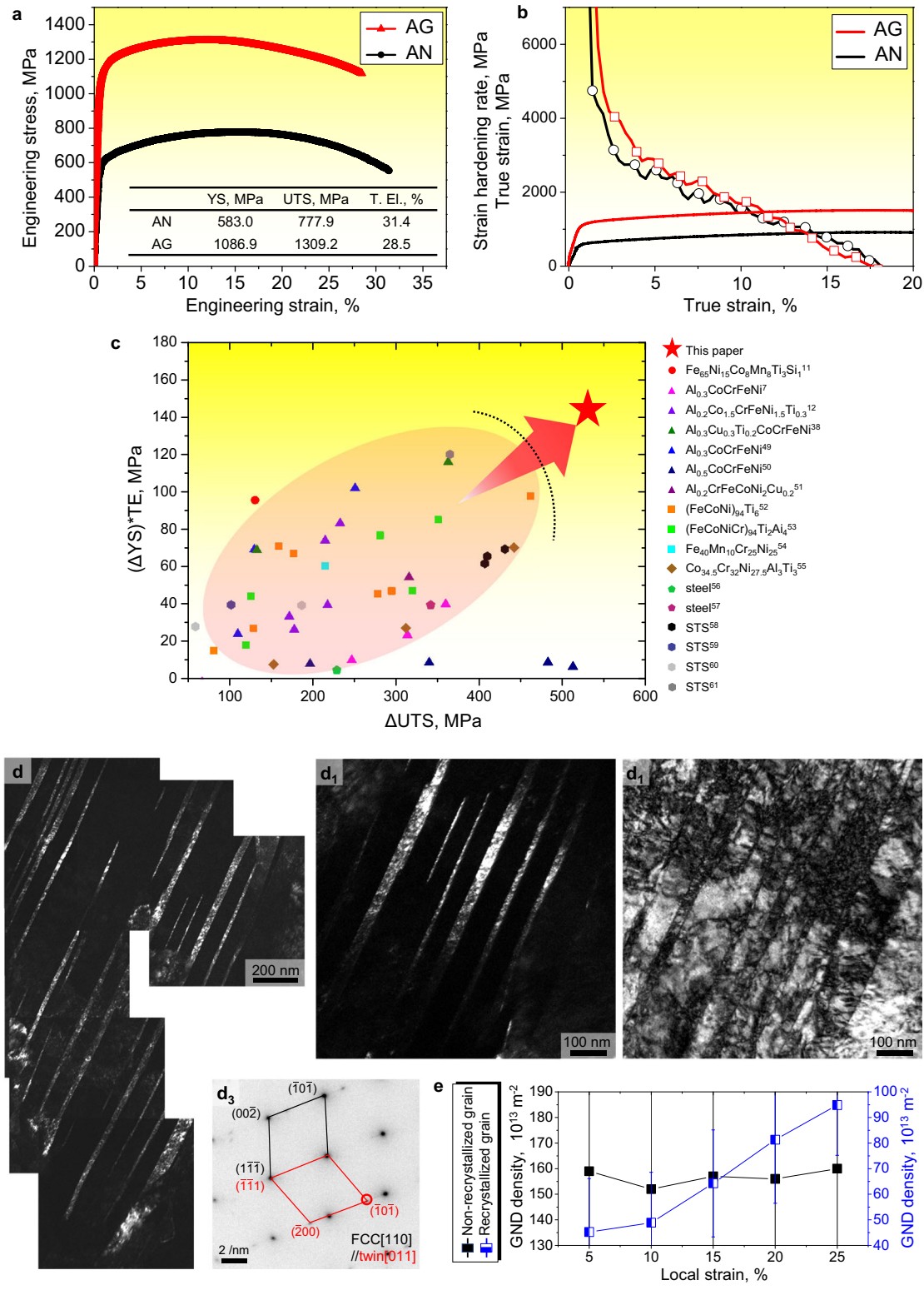

**Fig. 5 | Tensile properties and deformed microstructure of the aged sample.** **a** Engineering stress-strain curves, **b** strain hardening rate versus true stress-strain curves of experimental samples, and **c** comparison of strengthening among the annealed and aged H/MEA samples and conventional steels reported in the literature[7,11,12,38,49–61]. Deformed aged sample at 5% local true strain: (**d**) merged TEM

dark-field image; (**d₁**) TEM dark-field image obtained from the red circle in (**d₃**); **d₂** TEM bright-field image; **d₃** SAED pattern of (**d₂**); **e** GND density of each true strain level obtained via EBSD analysis. "AN" and "AG" denote annealed and aged alloys, respectively.

image quality images from local true strain levels ($\varepsilon_{tr}$) of 15% and 25% are also shown in Supplementary Fig. 11. The twin boundary increased from 7.4% at the 15% $\varepsilon_{tr}$ to 21.0% at the 25% $\varepsilon_{tr}$. As the $\varepsilon_{tr}$ increases, deformation twins, initially confined to the non-recrystallized region,

appear in the recrystallized region. The small twin spacing reduces the potential for dislocation pile-up, requiring more external stress to surpass and propagate across the twin boundary, thus contributing to work hardening[36,37]. Therefore, the increase in twin boundary fraction

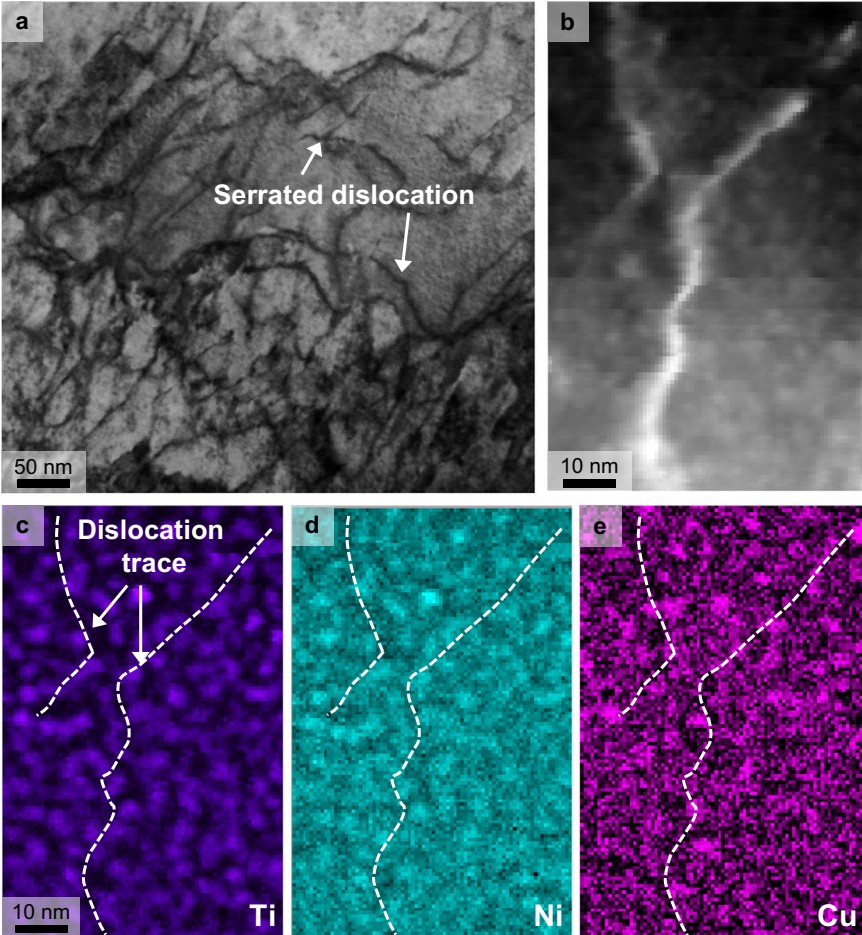

**Fig. 6 | The interaction of dislocation and nanoparticle under 2% strain. a** STEM bright-field image reveals the wavy and serrated dislocations. **b** STEM dark-field image and EELS spectrum maps of (**c**) Ti–$L_{2,3}$, **d** Ni–$L_{2,3}$, and (**e**) Cu–$L_{2,3}$.

indicates that twinning-induced plasticity is one of the main deformation mechanisms of the corresponding Fe–MEA. Phase transformation has not been detected upon the deformation due to the high stacking fault energy of the present FCC structure[38].

In addition, in the presence of a heterogeneous structure resulting from a partially recrystallized area, deformation incompatibility between the recrystallized and non-recrystallized regions generates GNDs and subsequently hetero deformation induced (HDI) strengthening as an additional strengthening mechanism. As shown in Fig. 5e, GND value development in the non-recrystallized and recrystallized regions as a strain was obtained from EBSD results of the aged sample with $\varepsilon_{tr}$ 5–25%; the accumulation of GNDs occurred predominantly in the recrystallized grains, leading to long-range internal stress, i.e., back stress[39]. The number of GNDs in the recrystallized grains increased continuously until they reached a strength level comparable to that of the non-recrystallized grains, hindering dislocation motion and developing long-range back stresses for HDI strengthening and strain hardening[40–42]. Furthermore, precipitation strengthening through the Orowan mechanism—activated by variously reported precipitates in the size range of 20–200 nm[43]—is evident in the STEM and corresponding EDS images (Supplementary Fig. 12). However, its contribution to the yield strength enhancement of 503 MPa was only ~138 MPa. An additional deformation mechanism is expected to further strengthen the aged sample and maintain its elongation.

### Spinodal hardening
The periodic spinodal structure distributed in the matrix induced a coherency strain in the microstructure, which arose from the

mismatch between the Fe– and Cu-rich FCC regions and the Ni(Ti, Si) B2 region. This lattice mismatch generated an internal stress distribution within the material. The distributed internal stress can inhibit the dislocation motion and enhance the deformation behavior and material strength at the macroscale[44]. As described in the "Methods" section, the strengthening contribution of the spinodal decomposed structure was quantified as ~327 MPa. Notably, according to the periodic internal stress, the spinodal strengthening mechanism contributed more significantly to the overall strengthening effect than precipitation strengthening, indicating that the internal stress effectively inhibited the dislocation motion without causing stress concentration and dislocation blocking owing to its high coherency, thus allowing ductility retention in the aged sample. Figure 6 shows that dislocation movement is hindered by the periodically decomposed nanoparticles, resulting in wavy and serrated dislocations at 2% local true strain. Figure 6a presents an overview of the deformed STEM bright-field image. Figure 6b–e demonstrates that the dislocation is impeded by the spinodal decomposed structure.

Therefore, spinodal decomposition—achievable using a proper alloying design strategy—can be considered an effective parameter for strengthening a material. Owing to the simple route for achieving precipitation and spinodal decomposition in a one-step process, this approach can easily be extended to other alloying systems to increase their mechanical strength without significantly sacrificing total elongation.

In conclusion, we demonstrated the efficiency of employing spinodal decomposition as a strategic approach for improving the strength-ductility trade-off in a developed Fe–MEA. Microstructural

observations revealed that the deliberate introduction of small amounts of constitutive elements, i.e., Cu and Al, provoked spinodal decomposition and the formation of a well-defined periodic spinodal architecture within the Fe− and Cu−rich FCC regions and the Ni(Ti, Si) B2 region. In addition, multiple dynamic precipitates nucleated during the aging process. The experimental results confirmed the success of the proposed approach. Compared with an annealed sample, an aged sample exhibited an enhancement of 187% in yield strength (1.1 GPa) while preserving 28.5% of the total elongation. The randomly distributed precipitates contributed a mere 138 MPa to the yield strength improvement, whereas the spinodal decomposed structure, with its periodic chemical fluctuations, contributed significantly to the strength (327 MPa) and elongation preservation. This study expands our understanding of the spinodal hardening principles and demonstrates their potential as an advanced paradigm in alloy design strategies, thus providing the foundation for further development of advanced metallic materials with improved performance.

## Methods

### Alloy fabrication

An ingot with a chemical composition of $Fe_{61.75}Ni_{14.25}Co_{7.6}Mn_{7.6}$ $Ti_{2.85}Si_{0.95}Cu_{4.5}Al_{0.5}$ (at%) was fabricated via vacuum induction melting (VIM, Induthern, Germany) using pure metals (>99.99%) in high purity Ar gas inside a graphite mold with dimensions of $7 \times 33 \times 80\ mm^3$. The 150 g as cast ingot was homogenized at 1150 °C for 6 h in an Ar atmosphere, followed by cold rolling from 7−1.5 mm (78.5% thickness reduction). Subsequently, the cold rolled sample was annealed at 850 °C for 5 min in an Ar atmosphere and then water quenched to obtain a partially recrystallized microstructure. The annealed specimen was aged at 550 °C for 5 h and then water quenched. Computational thermodynamics calculations were employed to predict the outcome of our design strategy, the equilibrium phase fraction versus temperature calculated by Thermo−Calc 3.0 with TCHEA database.

### Mechanical−property tests

A flat dogbone-shaped sample with a thickness of 1.5 mm, width of 2.5 mm, and a gauge length of 5.0 mm was cut along the rolling direction. A room temperature tensile test utilizing digital image correlation (ARAMIS M12, GOM Optical Measuring Techniques, Germany) was conducted using a universal testing machine (Instron 1361, Instron Corp., USA) at a strain rate of $1 \times 10^{-3}\ s^{-1}$. To ensure reproducibility, tensile tests were conducted at least three times for each condition.

### Microstructure characterization

The initial and deformed microstructures were characterized via EBSD (XL30S FEG, Philips and JSM−7100, JEOL, Japan) and field emission scanning electron microscopy. The annealed, aged, magnified annealed, and deformed microstructure was obtained through EBSD measurement using a step size of 1.0 μm, 1.0 μm, 0.15 μm, and 0.2 μm, respectively. The results were analyzed using orientation imaging microscopy collection software (TSL OIM Analysis 7). To quantify the relative change in GND density from EBSD measurement, strain gradient theory[45,46] was applied as follows:

$$\rho_{GND} = \frac{2\theta}{ub},\quad (1)$$

where $u$ is the step size (0.2 μm), $b$ is the magnitude of the Burgers vector, and $\theta$ is the average local misorientation angle measured in KAM maps.

The aged specimens for TEM analysis were prepared via mechanical polishing to a thickness of 70 μm and electropolishing in a solution of 10% $HClO_4$ and 90% $CHCOOH$ at an applied voltage of 20 V at 25 °C. A deformed aged specimen with local true strain levels at 2% and 5% was prepared through the focused ion beam (FIB) lift-out

procedure (SII SMI3050SE, SII Nanotechnology, Japan). TEM was performed to examine the microstructure (JEM−2100 F, JEOL, Japan) at an accelerating voltage of 200 kV. The APT samples were prepared using dual beam FIB technology with a $Ga^+$ ion beam (FEI Helios Nanolab 650i). APT analysis was conducted using a LEAP 4000X HR instrument (CAMECA Inc), which allowed the observation of the elemental distribution within the aged sample. The APT data were analyzed using the commercial IVAS® software (ver. 3.8.10) by Cameca. For the analysis of the deformed microstructure, local true strain levels were obtained from DIC data (See Supplementary Fig. 13).

### Synchrotron XRD

Synchrotron XRD measurements were performed on the aged samples at 550 °C for 1, 2, and 5 h at the 8D beamline of Pohang Accelerator Laboratory with a X-ray energy of 12.0 keV and wavelength of 0.10258 nm. To observe the shift of the sideband, the scan results for three aged samples were obtained in the 2θ range of 26−30° with steps of 0.01° and a holding time of 0.5 s per step.

### Precipitation strengthening

Precipitation strengthening involves two primary mechanisms: Orowan bypass and shearing. The Orowan bypass mechanism refers to the interaction between dislocations and precipitates, where dislocations bow around large precipitates, impeding their motion and contributing to the overall strengthening[47]. This mechanism can be quantified by the following equations[11]:

$$\Delta\sigma_{ppt} = M\frac{0.4Gb}{\pi\lambda}\frac{\ln\frac{2\bar{r}}{b}}{\sqrt{1-v}},\quad (2)$$

$$\lambda = \sqrt{\frac{4}{3}}\bar{r}\sqrt{\frac{\pi}{4f}-1},\quad (3)$$

where $M$ represents the Taylor factor for the FCC matrix (~3.06), $G$ represents the shear modulus (~78.4 GPa was assumed in this study[11]), $b$ is the burgers vector (0.2548 nm, measured from the XRD data), and $v$ is the Poisson's ratio (0.3). $\lambda$ represents the inter-precipitate spacing, $\bar{r}$ represents the mean precipitate radius, and $f$ is the volume fraction of the precipitate.

### Spinodal hardening

Spinodal hardening results from spinodal decomposed structures with a coherent interface and fluctuations in the chemical composition[15]. Kato et al.[25] developed the hardening mechanism in spinodal decomposition and quantified the increase in yield strength due to coherency stress in a spinodal modulated structure in FCC alloys by considering the dislocation force balance equation. The resistance to the dislocation motion increased proportionally to the amplitude of the composition gradient, regardless of the fluctuation wavelength. The equation proposed by Kato et al.[25] to quantify the strengthening induced by spinodal decomposition in FCC alloys is

$$\sigma_{sd} = \frac{A\eta Y}{\sqrt{6}},\quad (4)$$

where $A$ represents the amplitude of chemical composition in the spinodal decomposed structure in atomic percent (calculated as 0.5% for Cu); η represents lattice mismatch $((1/a)(\partial a/\partial c))$, which is the composition variation of stress−free lattice parameter a concerning the Cu concentration (0.00752 as calculated from Supplementary Fig. 8); $Y$ represents elastic modulus calculated from elastic constant $C_{ij}$. When $2C_{44} - C_{11} + C_{12} > 0$, the elastic energy is minimal, and spinodal decomposition develops in the elastically soft direction of $[001]$[25]. $Y_{(100)}$ is calculated as $Y_{(100)} = (C_{11} + 2C_{12})(C_{11} - C_{12})/C_{11}$ and

estimated as 219.98 GPa from $C_{11} = 271$ GPa and $C_{12} = 175$ GPa[48]. Therefore, the spinodal strengthening is calculated as 327 MPa, using $A = 0.5\%$, $\eta = 0.00752$, and $Y = 219.98$ GPa.

## Data availability

The data generated during and/or analyzed during the current study are available from the corresponding author upon reasonable request.

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

## Acknowledgements

Dr. Haftlang was supported by the National Research Foundation of Korea (NRF) grant funded by the Ministry of Science and ICT of Korea (2021R1A2C3006662). This research was supported by the Nano & Material Technology Development Program through the National Research Foundation of Korea (NRF) funded by the Ministry of Science and ICT (RS-2023-00281246). This work was supported by the National Research Foundation of Korea (NRF) grant funded by the Korean government (MSIT) (NRF-2022R1A5A1030054). The authors appreciate the Pohang Accelerator Laboratory (Pohang, Republic of Korea) for providing the synchrotron radiation sources at the 8D beamlines used in this study. The first author kindly thanks Prof. Sun Ig Hong for the helpful discussion.

## Author contributions

H. Park and F. Haftlang conceived the study. F. Haftlang designed the alloy. H. Park prepared the materials, performed tensile tests, and conducted structural characterizations. Y.-U. Heo performed the TEM analysis. J.B. Seol contributed to the APT reconstruction. F. Haftlang conducted the APT reconstruction and analysis. Zhijun Wang contributed microstructural analysis. H. Park, F. Haftlang, Y.-U. Heo, and H.S. Kim wrote the manuscript. All the authors contributed to the discussion of the results.

## Competing interests

The authors declare no competing interests.
