## [Peer Review File · Nature Communications]

Periodic spinodal decomposition in double–strengthened medium–entropy alloyREVIEWER COMMENTS

Reviewer #1 (Remarks to the Author):

Key results

The submitted manuscript suggests using matrix hardening through spinodal decomposition, inferring a level of strength gain with no impediment to toughness, as opposed to mere precipitation strengthening. Authors confidently use a large complimentary set of experimental techniques to address this research question, based on theoretical assumption made by other teams.

Validity

The presented work does a good job of disseminating collected datasets. The measured elongation increase of ~28% shows that - to some extent - the proposed retention/increase of mechanical properties can be achieved.

However, some comments remain:

- Fig. 1 and associated text do not show comparable datasets for annealed and aged state, which makes it more difficult for the reader to follow argumentation about proposed mechanisms.
- Ext. figure 6b, showing the crystallographic information for area imaged in 6a, is hard to interpret due to its high contrast and poor image quality; a higher resolution broad beam SAED pattern of representative region could be helpful
- A more in-depth explanation of the origins of the purported core-shell structure seen in the APT images is advised
- The insert image in Fig.2e is rather small, the axes text should be increased for readability; further the axis notation varies from at-pct to a.u., though it should be the same unit?

Data and methodology

Generally, significant efforts were undertaken in support of this work, involving a wide range of characterization techniques, as well as mechanical testing.

- Induction melting in graphite mold involves the potential for significant carbon pickup, which should be a point of consideration in a Fe-based alloy system
- How were GND measurements conducted?
- How many mechanical tests were conducted, including replicates? This is crucial to know for assessing statistical significance of results
- A further question is the originality aspect of the alloy design; there is little mention of how the authors arrived at the proposed composition, whether this was informed by statistical and computational modeling. Related, the employed thermal annealing processing steps are not referenced by prior work or open literature, specifically the aging temperature leading to successful spinodal decomposition

Reviewer #2 (Remarks to the Author):

Comments on "Periodic spinodal decomposition in double-strengthened medium-entropy alloy".

This paper presents significant findings on the remarkable strengthening effects observed in a ferrous medium-entropy alloy (Fe_{61.75}Ni_{14.25}Co_{7.6}Mn_{7.6}Ti_{2.85}Si_{0.95}Cu_{4.5}Al_{0.5}), with notable attention given to the achievement of a high tensile stress value. The feedback provided by the reviewer has been presented, and addressing these points properly will contribute to the paper's overall quality and its potential acceptance for publication.

1. GND Map in Fig. 1b:

While conventional EBSD measurements have been employed to explore the GND density, the reviewer raises concerns about its accuracy. To improve the precision of GND density determination, it is recommended to conduct a crosscourt analysis. It is preferably utilizing the KAM profile here.

2. Analysis of the Moire Pattern in Fig. 2d:

The Moire pattern analysis in Fig. 2d has been questioned due to the observed extension of the pattern beyond the expected particle size of approximately 5nm. This suggests the possibility of Moire pattern generation from the overlap of two grains within the TEM sample, potentially due to excessive sample thickness. To address this concern, a discussion on the influence of sample thickness on Moire pattern observations is necessary.

Additionally, the authors should clarify the schematic diagram presented in Fig. 2f. A detailed explanation of the intended information conveyed by the middle figure of Fig. 2f will enhance the reader's comprehension.

3. Strengthening Mechanism from Spinodal Decomposition:

The authors have attributed the major contribution to strengthening to spinodal decomposition. To support this claim, it is crucial to provide evidence of the interaction between dislocations and the decomposed nano-particles from the TEM observation.

The presence of twin structures after deformation raises questions regarding the potential contribution of twinning-induced plasticity to strengthening. The authors should provide a clear explanation of why they believe the strengthening mechanism is not primarily driven by twinning and offer supporting data or reasoning.

Lastly, clarify the methodology behind plotting Fig. 3c, particularly considering that the TEM sample is from a 5% locally true strained sample. A detailed description of how this figure was generated will aid in the reader's understanding.

Addressing these points comprehensively and providing the necessary evidence and explanations will significantly enhance the paper's quality and its potential for acceptance in the publication.

AUTHORS' POINT-BY-POINT RESPONSE TO THE REVIEWERS' COMMENTS
Nature Communications

Ref. No.: NCOMMS-23-63788-T

Dear Reviewers,

Thank you very much for your assessment and constructive comments on our manuscript (NCOMMS-23-63788-T) entitled *Periodic spinodal decomposition in double-strengthened medium-entropy alloy*. We realized that our previous manuscript had some unclear points. We have addressed these issues in the revised manuscript. In the revised manuscript, the responses to the reviewers are indicated in **blue**, and the revised parts were highlighted in **yellow**.

We hope that our revisions are satisfactory and clear.

With best regards on behalf of the author team,

Hyung Seop Kim

Response to reviewer #1:

Key results

The submitted manuscript suggests using matrix hardening through spinodal decomposition, inferring a level of strength gain with no impediment to toughness, as opposed to mere precipitation strengthening. Authors confidently use a large complimentary set of experimental techniques to address this research question, based on theoretical assumption made by other teams.

Comment #1:

Validity

The presented work does a good job of disseminating collected datasets. The measured elongation increase of ~28% shows that - to some extent - the proposed retention/increase of mechanical properties can be achieved.

However, some comments remain:

- Fig. 1 and associated text do not show comparable datasets for annealed and aged state, which makes it more difficult for the reader to follow argumentation about proposed mechanisms.

Response #1:

We are grateful for this constructive comment. We have added the EBSD image of the aged sample in **Supplementary Figure 2** to elucidate the microstructural evolution upon the aging process and improve the manuscript by providing further explanation. Please see page 4 lines 97 to 100:

Figures 1a and **b** show the electron backscatter diffraction (EBSD) phase and inverse pole figure (IPF) maps of the annealed sample. The phase of this sample was nearly fully FCC, with a small amount of martensite thermally induced by water quenching. Additionally, it

exhibited a partially recrystallized microstructure with fine grains and an average size of ~ 4.35 μm . The volume fraction of the recrystallized grains was approximately 30%, with predominant nucleation along the shear bands²¹ and grain boundaries²². The microstructure of the aged samples is shown in **Supplementary Figure 2**. Notably, it reveals that the annealed and aged samples have similar microstructures on the microscale because the aging temperature of 550 $^{\circ}\text{C}$ is not high enough to induce recrystallization.

Supplementary Figure 2. Aged sample. EBSD a phase and b IFP maps of the aged sample.

Comment #2:

Ext. figure 6b, showing the crystallographic information for area imaged in 6a, is hard to interpret due to its high contrast and poor image quality; a higher resolution broad beam SAED pattern of representative region could be helpful.

Response #2:

Thank you for your comment on Extended Fig. 6b. We aimed to capture the wavelength of the spinodal decomposition by analyzing the chemical composition along the $[001]_{\text{FCC}}$ direction, which exhibits a strong preferential direction for spinodal decomposition. For this purpose, we utilized the image in Extended Figure 6b, derived from atom probe tomography (APT) data, with the orientation information reprocessed using field ion microscopy. Regrettably, the corresponding image is poor compared to the electron backscatter diffraction (EBSD) pattern due to its origin from extracted ions in the APT data. In response to your suggestion, we have redrawn the image for better clarity, as shown in **Figure 3b**. This figure illustrates the hit detector map of the region delineated in **Figure 3a**, elucidating the crystallographic poles. The protocol described in the previous literature¹ was employed to obtain this image. We trust that this improved representation aids in your interpretation of the image and understanding of our purpose.

Figure 3. APT reconstruction of the aged sample. a APT reconstructed maps of alloying species in the studied alloy, showing the modulated structure. **b** Hit detector map of the region marked by a black box, oriented along the $[001]$ direction. Crystallographic main poles are indexed on the map. **c** One-dimensional concentration profile along the $[001]_{\text{FCC}}$ and

normalized composition profiles for Fe and Cu, with the maximum value of each element set to 1 for comparison in **c**.

1. Breen, A., Day, A., Lim, B., Davids, W. & Ringer, S. Revealing latent pole and zone line information in atom probe detector maps using crystallographically correlated metrics. *Ultramicroscopy* **243**, 113640 (2023).

Comment #3:

- A more in-depth explanation of the origins of the purported core-shell structure seen in the APT images is advised

Response #3:

We thank the reviewer for the feedback. The formation process of the core-shell structure unfolds as follows: During aging, the negative enthalpy of mixing between Ni and Ti drives the formation of a Ni-Ti-rich region. The composition of Ni is more than four times higher than that of Ti in this alloy. Most Ti combines with Ni to form the Ni-Ti-rich region, occupying all available vacancies. Consequently, the excess Ni, which tends to interact with Ti to form a unified structure with Ni-Ti bonding, aggregates around the Ni-Ti bonding to constitute the shell. However, the formation kinetics of the core-shell microstructure and nanoprecipitates in detail are beyond the scope of the current study. Please see page 7 lines 142 to 150:

“Based on the reconstructed APT interpretation, the isosurface maps for the two elements are shown in **Figure 2a-c**, with Ti and Ni strongly tangled with each other. Upon closer inspection, a core-shell structure was observed, with Ni covering Ti, as shown in **Figure 2a2**. The formation of the core-shell structure is influenced by the mixing enthalpy of the constituent elements. During aging, the negative enthalpy of mixing between Ni and Ti drives the formation of a Ni-Ti-rich region. The atomic fraction of Ni is more than 4 times higher

than that of Ti in this alloy. Most of the Ti combines with Ni to form the Ni–Ti– rich region, occupying all available vacancies. Consequently, the excess Ni, which tends to interact with Ti to form a uniform structure, aggregates around the Ni–Ti bonding to constitute the shell. The core–shell structure likely resulted from uphill diffusion, thus providing indirect evidence for spinodal decomposition²⁸. Though, the formation kinetics of the Ni–Ti core–shell nanoprecipitates in detail are beyond the scope of the current research.”

Comment #4:

- The insert image in Fig.2e is rather small, the axes text should be increased for readability; further the axis notation varies from at-pct to a.u., though it should be the same unit?

Response #4:

We appreciate the feedback regarding the insert image in Figure 2e. In response to your suggestions, we have taken measures to enhance the readability of the figure. Specifically, we have increased the font size of the axis text to ensure that it is easily legible.

To enhance the illustration of chemical composition fluctuation, particularly between Cu and Fe within the alloy system, we have added a separate subplot within Figure 3c that specifically emphasizes the fluctuation of Cu and Fe compositions. This addition aims to provide a more precise visualization of their distinct sinusoidal pattern. To facilitate a direct comparison between these compositions, we normalized their maximum values to 1. Therefore, the Y–axis is normalized intensity, arbitrary units (a.u.) are noted to represent these normalized values. Furthermore, we have enriched the figure legend with additional explanations concerning the normalization as follows:

Figure 3. APT reconstruction of the aged sample. **a** APT reconstructed maps of alloying species in the studied alloy, showing the modulated structure. **b** Hit detector map of the region marked by black box, oriented along the [001] direction. Crystallographic main poles are indexed on the map. **c** One-dimensional concentration profile along the [001]_{FCC} and Normalized composition profiles for Fe and Cu, with the maximum value of each element set to 1 for comparison in **c**.

Comment #5:

Data and methodology

Generally, significant efforts were undertaken in support of this work, involving a wide range of characterization techniques, as well as mechanical testing.

- Induction melting in graphite mold involves the potential for significant carbon pickup, which should be a point of consideration in a Fe-based alloy system

Response #5:

As the reviewer has concerned, the carbon concentration in Fe-based alloys is an important factor in microstructure and mechanical properties, including the stability of the FCC matrix and the formation of precipitate phases. We have conducted carbon/sulfur determination using (LECO/CS844 C/S Analyzer, Leco Corporation, St. Joseph, USA) to analyze carbon content. In our Fe-based alloy, the carbon concentration of 107 ± 5.6 ppm is closely aligned with the levels found in extremely low carbon steels². This specific concentration is below the threshold for influencing the formation of precipitating phases and carbides or changing phase stability that could significantly alter material properties. This minimal carbon level ensures that the primary effects of carbon, such as phase stabilization and minor adjustments to mechanical properties, do not lead to the formation of carbides or other precipitates that could detrimentally affect the alloy's performance.

2. Koga, N., Kanehira, Y., Huyen, P.T.T., Hori, K. & Umezawa, O. Effect of aging on low-temperature tensile properties of ultra-low carbon steel. *ISIJ International* **61**, 2308-2316 (2021).

Comment #6:

- How were GND measurements conducted?

Response #6:

Thank you for the comment. We calculated the GND from the KAM values obtained from EBSD data (presented in **Figure 5c**). We provide further details on how we measured the GND value in the Method section. Please see page 19 lines 345 to 353:

The annealed, aged, magnified annealed, and deformed microstructure were obtained through EBSD measurement using a step size of 1.0 μm , 1.0 μm , 0.15 μm , and 0.2 μm , respectively. The results were analyzed using orientation imaging microscopy collection software (TSL OIM Analysis 7). To quantify the relative change in GND density from EBSD measurement, strain gradient theory^{3,4} was applied as shown in Equation (1):

$$\rho_{\text{GND}} = \frac{2\theta}{ub}, \quad (1)$$

where u is the step size (0.2 μm), b is the magnitude of the Burgers vector, and θ is the average local misorientation angle measured in KAM maps.

3. Gao, H., Huang, Y., Nix, W.D. & Hutchinson, J.W. Mechanism-based strain gradient plasticity - I. Theory. *J Mech Phys Solids* **47**, 1239-1263 (1999).
4. Kubin, L.P. & Mortensen, A. Geometrically necessary dislocations and strain-gradient plasticity: a few critical issues. *Scripta Materialia* **48**, 119-125 (2003).

Comment #7:

- How many mechanical tests were conducted, including replicates? This is crucial to know for assessing statistical significance of results.

Response #7:

We thank the reviewer for this constructive comment. To ensure reproducibility, we performed the tensile test at least three times for each condition. The stress-strain curves of the reproducibility test are presented in Figure R1. While minor variations were observed in the reproducibility experiments, they remained within acceptable limits. The significant improvement in strength observed between the annealed and aged samples in every condition cannot solely be attributed to precipitation; it is also influenced by the spinodal decomposed structure described in our manuscript. We have added a note about reproducibility in the Method section. Please see page 19 lines 341 to 342:

Mechanical-property tests

A flat dog-bone-shaped sample with a thickness of 1.5 mm, width of 2.5 mm, and gauge length of 5.0 mm was cut along the rolling direction. A room-temperature tensile test utilizing digital image correlation (ARAMIS M12, GOM Optical Measuring Techniques, Germany) was conducted using a universal testing machine (Instron 1361, Instron Corp., USA) at a strain rate of $1 \times 10^{-3} \text{ s}^{-1}$. To ensure reproducibility, tensile tests were conducted at least three times for each condition.

Figure R1. **a** engineering stress–strain curve of annealed samples and **b** engineering stress–strain curve of aged samples.

Comment #8:

- A further question is the originality aspect of the alloy design; there is little mention of how the authors arrived at the proposed composition, whether this was informed by statistical and computational modeling.

Related, the employed thermal annealing processing steps are not referenced by prior work or open literature, specifically the aging temperature leading to successful spinodal decomposition.

Response #8:

We appreciate the reviewer's insightful inquiries concerning the originality of our alloy design and the specifics of the heat treatment process. Addressing the first point on alloy composition, our selection of the complex elemental alloy system is grounded in thermodynamic principles. The thermodynamic principle, $\Delta G = \Delta H - T\Delta S$, where ΔG represents Gibbs free energy, ΔH is the mixing enthalpy, ΔS is the mixing entropy, and T denotes temperature. The introduction of a large number of constituent elements, while beneficial for increasing the entropy, also has the potential to increase the mixing enthalpy⁵. The current Fe_{61.75}Ni_{14.25}Co_{7.6}Mn_{7.6}Ti_{2.85}Si_{0.95}Cu_{4.5}Al_{0.5} MEA alloy has been designed as a branch of the Fe–Ti–Si–based nanoprecipitation–strengthened medium–entropy alloys inspired by the previous alloys Fe₆₅Ni₁₅Mn₈Co₈Ti₃Si₁⁶ and Fe₆₈Ni₁₀Mn₁₀Co₁₀Ti_{1.5}Si_{0.5}⁷. An extraordinarily high mechanical strength has been reported after the formation of Fe₂SiTi and Ni₃Ti nanoprecipitates, yet high ductility as the result of the presence of non–shearable precipitates^{6–14}. On the other hand, the strong ordering tendencies of Al, along with most of the other alloying elements, make the ordering transition of primary importance. A sufficient concentration of Al encouraged a series of phase transformations, including the formation of coherent mixtures of B2 and partially ordered aluminides in a body–centered cubic (BCC) solid–solution matrix^{15–17}. In the presence of Al–containing B2 or BCC phase, the mechanical strength incredibly increases at the expense of ductility. Therefore, to avoid these phase transformations leading to ductility reduction, yet using the advantage of lattice expansion and solid solution strengthening, a low concentration of 0.5 at% was chosen for the Al element. In the Fe–based alloy and CrMnFeCoNi alloy systems, previous literature has revealed microstructural change with the addition of Cu^{18–21}. The addition of Cu above 5% can induce Cu–rich precipitation or microscale phase separation²¹. Therefore, in our study, we have strategically limited the Cu addition to below 5% to induce the desired nanostructures without promoting excessive phase separation.

By controlling the composition and microstructure through thermal processing associated with the phase transformations, this system provides a wide range of potential compounds and

associated microstructures. Thus, the main concept of the present work is to design a stable FCC matrix to enhance its sensitivity to spinodal decomposition. Based on the thermodynamic calculations and the equilibrium phase diagram presented in **Supplementary Figure 1**, the presence of Fe₂SiTi, Ni₃Ti, B2, and Cu-rich FCC phases is possible upon a suitable heat treatment in the range of ~450–600 °C. Therefore, by applying the similar aging treatment we have done in our previous work (at 550 °C for 5 hours⁶, as shown in the differential scanning calorimetry (DSC) analysis in **Supplementary Figure 14**), all the aforementioned precipitates have been formed in our microstructure. This results in an excellent combination of UTS and uniform elongation compared to all these reported alloys. Besides, the spectacular characteristic of the core-shell microstructure classifies this Fe-MEA as an alloy with high potential for high-strength applications.

Therefore, such a complex multi-heterogeneous microstructure of the current Fe-MEA is apparently distinct from the simpler microstructure of the previous one⁶. The current manuscript also intensifies the role of alloying elements on the phase stability of the initial microstructure, showing how the microstructure converts from a solid solution state to a precipitate/spinodal decomposed structure strengthened one with only a small change in compositional elements compared to the previous Fe₆₅Ni₁₅Mn₈Co₈Ti₃Si₁ MEA⁶.

We have revised the **introduction** based on this information and added it as **Supplementary Note 1. Alloy Design Strategy**, as follows:

Introduction

To realize spinodal decomposition in multi-principal element alloys, we designed a new ferrous medium-entropy alloy (Fe_{61.75}Ni_{14.25}Co_{7.6}Mn_{7.6}Ti_{2.85}Si_{0.95}Cu_{4.5}Al_{0.5} denoted as Fe-MEA) by adding the minor elements Cu and Al. This deliberate compositional modification introduces the formation of a miscibility gap and subsequent spinodal decomposition. **This complex compositional strategy not only increases the entropy of the alloy but also enhances its enthalpy, thereby improving the probability of spinodal decomposition¹⁸. Cu, which has a large miscible gap with Fe—the principal component of the alloy—was added 4.5%. Adding over 5% of Cu is avoided due to the formation of microscale phase separation and immoderate Cu-rich precipitates¹⁹. Al was deliberately selected at a concentration of 0.5 at% to strategically leverage the benefits of lattice expansion and solid solution strengthening. Both Cu and Al are**

widely known to increase the stacking fault energy, preventing the generation of transformation-induced plasticity²⁰. Based on the phase diagram (Supplementary Figure 1), the presence of Fe₂SiTi, Ni₃Ti, B2, and Cu-rich FCC phases is possible upon a suitable heat treatment in the range of ~450–600 °C; thus the appropriate heat treatment should be performed to achieve the desired spinodal decomposition and the formation of multiple dynamic precipitates simultaneously (detail describes in Supplementary Note 1 and method section). Consequently, Fe–MEA utilizing spinodal decomposition shows a doubled strength without notable ductility loss. Leveraging spinodal decomposition holds significant promise for advancing alloy design strategies and overcoming the long-standing challenge of balancing strength and ductility in H/MEAs, opening up new horizons for developing next-generation metallic materials.

Supplementary Note 1. Alloy design strategy

The design of this alloy strategically exploits the diversity of constituent elements and their enthalpy contributions to promote spinodal decomposition. The thermodynamic principle, $\Delta G = \Delta H - T\Delta S$, where ΔG represents Gibbs free energy, ΔH is the mixing enthalpy, ΔS is the mixing entropy, and T denotes temperature, describes the significance of the high entropy value in M/HEAs. This high entropy stabilizes the Gibbs free energy, promoting a solid solution state in M/HEAs. Contrary to the beneficial "high entropy effect," an increased number of constituent elements can elevate the enthalpy, thereby enhancing the likelihood of spinodal decomposition².

The current Fe_{61.75}Ni_{14.25}Co_{7.6}Mn_{7.6}Ti_{2.85}Si_{0.95}Cu_{4.5}Al_{0.5} MEA alloy has been designed as a branch of Fe-Ti-Si-based nanoprecipitation-strengthened medium-entropy alloys inspired by the previous alloy Fe₆₅Ni₁₅Mn₈Co₈Ti₃Si₁³ and Fe₆₈Ni₁₀Mn₁₀Co₁₀Ti_{1.5}Si_{0.5}⁴. An extraordinarily high mechanical strength has been reported after the formation of Fe₂SiTi and Ni₃Ti nanoprecipitates, yet high ductility resulting from the presence of non-shearable precipitates³⁻¹¹. On the other hand, the strong ordering tendencies of Al, along with most of the other alloying elements, make the ordering transition of primary importance. A sufficient concentration of Al encouraged a series of phase transformations, including the formation of coherent mixtures of B2 and partially ordered aluminides in a body-centered cubic (BCC)

solid–solution matrix¹²⁻¹⁴. In the presence of Al–containing B2 or BCC phase, the mechanical strength incredibly increases at the expense of ductility. Therefore, to avoid these phase transformations leading to ductility reduction, while using the advantage of lattice expansion and solid solution strengthening, a low concentration of 0.5% was chosen for the Al element. The Fe–Cu binary system is widely known for its positive mixing enthalpy, which significantly tends towards phase separation. In the Fe–based alloy and CrMnFeCoNi alloy systems, previous literature has revealed microstructural changes with the addition of Cu¹⁵⁻¹⁸. The addition of Cu above 5 at% can induce Cu–rich precipitation or microscale phase separation¹⁸. Therefore, in our study, we have strategically limited the Cu addition to below 5% to induce the desired nanostructures without promoting excessive phase separation.

By controlling the composition and microstructure through thermal processing associated with the phase transformations, this system provides a wide range of potential compounds and associated microstructures. Thus, the main concept of the present work is to design a stable FCC matrix to enhance its sensitivity to spinodal decomposition. Based on the thermodynamic calculations and the equilibrium phase diagram presented in Supplementary Figure 1, the presence of Fe₂SiTi, Ni₃Ti, B2, and Cu–rich FCC phases is possible upon a suitable heat treatment in the range of ~450–600 °C. Therefore, by applying a similar aging treatment as we have done in our previous work³ (at 550 °C for 5 hours), as shown in the differential scanning calorimetry (DSC) analysis in Supplementary Figure 14, all the aforementioned precipitates have been formed in our microstructure.

5. Luan, H.W. *et al.* Spinodal decomposition and the pseudo-binary decomposition in high-entropy alloys. *Acta Mater.* **248** (2023).
6. Fillon, A. *et al.* Influence of severe plastic deformation on the precipitation hardening of a FeSiTi steel. *J. Mater. Sci.* **47**, 7939-7945 (2012).
7. Perrier, M., Bouaziz, O., Brechet, Y., Deschamps, A. & Donnadieu, P. Mechanical properties of low carbon steel hardened by the Fe₂SiTi phase at high volume fraction. *J. Phys.: Conf. Ser.* **240**, 012095 (2010).
8. Knowles, A.J. *et al.* Development of Ni-free Mn-stabilised maraging steels using Fe₂SiTi precipitates. *Acta Mater.* **174**, 260-270 (2019).

9. Perrier, M. *et al.* Precipitation sequence and kinetics in an Fe-Si-Ti alloy. *Solid State Phenom.* **172**, 833-838 (2011).
10. Perrier, M. *et al.* Characterization and modeling of precipitation kinetics in a Fe-Si-Ti alloy. *Metall. Mater. Trans. A* **43**, 4999-5008 (2012).
11. Löffler, F., Palm, M. & Sauthoff, G. Iron-Rich Iron-Titanium-Silicon Alloys with Strengthening Intermetallic Laves Phase Precipitates. *Steel Res. Int.* **75**, 766-772 (2004).
12. Jack, D. & Honeycombe, R. Age hardening of an Fe-Ti-Si alloy. *Acta Metall.* **20**, 787-796 (1972).
13. Haftlang, F., Seol, J.B., Zargarani, A., Moon, J. & Kim, H.S. Chemical core-shell metastability-induced large ductility in medium-entropy maraging and reversion alloys. *Acta Mater.* **256**, 119115 (2023).
14. Haftlang, F. *et al.* Simultaneous effects of deformation-induced plasticity and precipitation hardening in metastable non-equiatomic FeNiCoMnTiSi ferrous medium-entropy alloy at room and liquid nitrogen temperatures. *Scr. Mater.* **202** (2021).
15. Santodonato, L.J. *et al.* Deviation from high-entropy configurations in the atomic distributions of a multi-principal-element alloy. *Nat. Commun* **6**, 5964 (2015).
16. Singh, S., Wanderka, N., Murty, B., Glatzel, U. & Banhart, J. Decomposition in multi-component AlCoCrCuFeNi high-entropy alloy. *Acta Mater.* **59**, 182-190 (2011).
17. Santodonato, L.J., Liaw, P.K., Unocic, R.R., Bei, H. & Morris, J.R. Predictive multiphase evolution in Al-containing high-entropy alloys. *Nat. Commun* **9**, 4520 (2018).
18. Ma, X. *et al.* Spinodal decomposition of precipitation hardening Fe-17Cr-4Ni-4Cu stainless steel at 475 C. *Materials and technology* **56**, 193–199-193–199 (2022).
19. Fiocchi, J., Mostaed, A., Coduri, M., Tuissi, A. & Casati, R. Development and characterization of a novel high entropy alloy strengthened through concurrent spinodal decomposition and precipitation. *J. Alloy. Compd.* **947**, 169706 (2023).
20. Zhang, Y. *et al.* Concurrence of spinodal decomposition and nano-phase precipitation in a multi-component AlCoCrCuFeNi high-entropy alloy. *J. Mater. Res. Technol* **8**, 726-736 (2019).

21. Du, C. *et al.* Effect of Cu on the strengthening and embrittlement of an FeCoNiCr-xCu HEA. *Mater. Sci. Eng. A* **832**, 142413 (2022).

Response to reviewer #2:

Comments on "Periodic spinodal decomposition in double-strengthened medium-entropy alloy".

This paper presents significant findings on the remarkable strengthening effects observed in a ferrous medium-entropy alloy ($\text{Fe}_{61.75}\text{Ni}_{14.25}\text{Co}_{7.6}\text{Mn}_{7.6}\text{Ti}_{2.85}\text{Si}_{0.95}\text{Cu}_{4.5}\text{Al}_{0.5}$), with notable attention given to the achievement of a high tensile stress value. The feedback provided by the reviewer has been presented, and addressing these points properly will contribute to the paper's overall quality and its potential acceptance for publication.

Comment #1:

1. GND Map in Fig. 1b:

While conventional EBSD measurements have been employed to explore the GND density, the reviewer raises concerns about its accuracy. To improve the precision of GND density determination, it is recommended to conduct a crosscourt analysis. It is preferably utilizing the KAM profile here.

Response #1:

We sincerely appreciate the reviewer's valuable feedback. Our intention in Figure 2b is to illustrate the difference in dislocation density between recrystallized and non-recrystallized regions. The reviewer's concern regarding the accuracy of presenting GND density measured from EBSD in a single image has been noted. Therefore, we have decided to replace the GND map with a KAM map in Figure 2b, thereby ensuring a more precise representation of the data. Additionally, we have made corresponding edits to the manuscript. Please see page 4 lines 96 to 97:

Figures 1a and **b** show the electron backscatter diffraction (EBSD) phase and inverse pole figure (IPF) maps of the annealed sample. The phase of this sample was nearly fully FCC,

with a small amount of martensite thermally induced by water quenching. Additionally, it exhibited a partially recrystallized microstructure with fine grains and an average size of $\sim 4.35 \mu\text{m}$. The volume fraction of the recrystallized grains was approximately 30%, with predominant nucleation along the shear bands²² and grain boundaries²³. The magnified images of the annealed sample in **Figures 1b₁** and **b₂** reveal a discernible difference in **kernel average misorientation (KAM) value** between the recrystallized and non-recrystallized regions.

Figure 1. Compositional heterogeneities of the Fe–MEA. EBSD **a** phase and **b** IPF maps of the aged samples. Magnified **b₁** IPF and **b₂** KAM maps of an aged sample. **c** TEM bright–field image of the initial microstructure of the aged sample, along with STEM images and corresponding EDS maps of the **c₁** recrystallized region and **c₂** non–recrystallized region magnified in **c**.

Comment #2:

2. Analysis of the Moire Pattern in Fig. 2d:

The Moire pattern analysis in Fig. 2d has been questioned due to the observed extension of the pattern beyond the expected particle size of approximately 5nm. This suggests the possibility of Moire pattern generation from the overlap of two grains within the TEM sample, potentially due to excessive sample thickness. To address this concern, a discussion on the influence of sample thickness on Moire pattern observations is necessary.

Response #2:

We appreciate the reviewer’s critical comment. Even though the formation of nano–sized precipitates is illustrated in **Figures 1c₁** and **c₂**, the specimen thickness effect on the obtained phase contrast image hinders the direct understanding of the precipitate formation behavior. We have performed additional TEM observations to clarify the specimen thickness–dependent Moiré pattern generation, as shown in **Supplementary Figure 10**. Two specimen thicknesses (44 nm and 54 nm) were selected for comparison. The specimen thickness was measured using the low–loss spectra in electron energy loss spectroscopy (EELS). The detailed method of measuring specimen thickness is given in the previous report²². The STEM bright–field image in **Supplementary Figure 10b**, where the thickness is about 44 nm, shows isolated and overlapped distributions of the nanoprecipitates. This tendency coincides with the EDS Ti map in **Supplementary Figure 10b₁**. The HRTEM image was obtained in the same area (**Supplementary Figure 10b₂**). The Moiré patterns caused by the overlap of B2 and matrix

reflect the projected distribution of B2 particles in the matrix: some particles are isolated, and others are overlapped. However, the nanoprecipitates mostly overlap in the thicker region (54 nm) (Supplementary Figures 10c and c₁). The Moiré patterns in this area reveal the overlapping of B2 particles in Supplementary Figures 10c₃. We have added this discussion to the revised manuscript.

Supplementary Figure 10. Specimen thickness–dependent Moiré patterns. a STEM bright–field image showing variation in specimen thickness; **a₁** EELS low–loss spectra corresponding the positions **b** and **c** in **a**. **b** high magnification image of the position **b** in **a**; **b₁** EDS maps of Ti and Cu; **b₂** HR image of the corresponding region. **c** high magnification image of the position **c** in **a**; **c₁** EDS maps of Ti and Cu; **c₂** HR image of the corresponding region.

22. Heo, Y.U. Comparative study on the specimen thickness measurement using EELS and CBED methods. *Appl Microsc* 50, 8 (2020).

Comment #3:

Additionally, the authors should clarify the schematic diagram presented in Fig. 2f. A detailed explanation of the intended information conveyed by the middle figure of Fig. 2f will enhance the reader's comprehension.

Response #3:

Thank you for the reviewer's comment about the schematic image in Figure 2f. Our aged samples exhibit a complex microstructure characterized not only by a spinodal decomposed structure but also by a variety of precipitate phases and compositional heterogeneities. These features undoubtedly contribute to precipitation hardening and influence the material's mechanical properties. However, they are not the primary focus of our investigation. Our paper centers on elucidating the role and mechanisms of spinodal hardening in enhancing the mechanical properties of our alloy system. Therefore, the additional precipitate phases and compositional heterogeneity in detail are discussed in the supplementary materials. The supplementary section is intended to offer interested readers further insights into the microstructural complexity of our aged samples, including a thorough analysis of the diverse precipitate types and their distribution, as well as the overall compositional variability. For a better understanding of our manuscript, we have added summarized explanations of precipitate phases and compositional heterogeneities in the manuscript. Please see page 5 lines 109 to 117:

The TEM/STEM and atom probe tomography (APT) images revealed several distinct compositional heterogeneities in the formation of precipitates with distinctive elemental distributions. This includes the discovery of Fe_2SiTi and Ni_3Ti nanoprecipitates, which are abundant precipitates in the alloy system, alongside the observation of $\text{Ni}_3(\text{Ti}, \text{Si})_2$, a metastable

phase prone to transformation under prolonged aging²⁴. Additionally, elemental segregation near dislocations and at grain boundaries was identified, highlighting the diffusion paths that favor precipitation growth. The formation of distinct phases and clusters, such as Fe-rich BCC, Cu-rich FCC precipitates, and Cu clusters—are explained in detail in Supplementary Figures 3–7 and Supplementary Note 2.

Supplementary Note 2. Multiple dynamic precipitates in the aged sample

To clarify the microstructural features of the aged sample, TEM and APT analyses were conducted to investigate the nanoscale microstructural evolution during aging. The STEM image and corresponding EDS maps revealed several distinct compositional heterogeneities in the form of precipitates with distinctive elemental distributions, as follows:

(i) The presence of a Fe₂SiTi precipitate with a hexagonal close-packed (HCP) structure (P63/mmc, $a = 4.77 \text{ \AA}$, $c = 7.74 \text{ \AA}$) in the Fe–Mn–Cu–depleted area was confirmed by the SAED pattern corresponding to the area indicated by the white dashed line in the STEM and TEM dark-field images (see Supplementary Figures 3a and a2). Additionally, the reconstructed APT concentration profile (Supplementary Figure 3b₁) of the Fe₂SiTi precipitate in (Supplementary Figure 3b) indicated high concentrations of Si and Ti across the precipitate.

(ii) In addition to Fe₂SiTi, as indicated by white arrows in the EDS map in Supplementary Figure 3a₁, the presence of Ni- and Ti-rich areas (indicated by red arrows in the Ni EDS map in Supplementary Figure 3a₁) suggested the formation of Ni₃Ti nanoprecipitates. A perspective illustration of the Ni₃Ti precipitate is shown by the green dashed line in the STEM and dark-field images in Supplementary Figures 4a and a₁; the Ni₃Ti precipitate is enriched in Ni and Ti and depleted in Fe, Cu, Mn, and Co, as confirmed by the corresponding EDS results in Supplementary Figure 4a₃. The FFT in Supplementary Figures 5a–a₂ identifies the Ni₃Ti precipitate as an η -D0₂₄ nanoprecipitate with an HCP structure (P63/mmc, $a = 5.074 \text{ \AA}$, and $c = 8.31 \text{ \AA}$) along the $[2\bar{1}\bar{1}0]_{\eta}$ zone axis, with the matrix identified as FCC along the $[011]$ zone axis. The Ni₃Ti precipitate exhibits semi-coherency with the FCC matrix in the Nishiyama–Wassermann orientation with $(1\bar{1}1)_{\text{FCC}} \parallel (0001)_{\eta}$ and

$[011]_{\text{FCC}} \parallel [2\bar{1}\bar{1}0]_{\eta}$ ¹⁹. Ni_3Ti and Fe_2SiTi precipitates were abundant in the alloys, typically with average sizes of 20–200 nm.

(iii) More interestingly, the $\text{Ni}_3(\text{Ti}, \text{Si})_2$ nanoprecipitate—indicated by the yellow arrow in the Ni map in **Supplementary Figure 3a₁**—was confirmed as HCP (P63/mmc, $a = 4.57 \text{ \AA}$, and $c = 7.96 \text{ \AA}$) along the $[01\bar{1}0]_{\text{HCP}}$ zone axis, as shown in **Supplementary Figures 5b** and **b₁**. $\text{Ni}_3(\text{Ti}, \text{Si})_2$ is a metastable phase commonly observed in Ni–Ti–based alloys^{20,21}. The metastable phase can coarsen and decompose at high aging temperatures with long aging times to form the stable phase Ni_3Ti ²¹.

(iv) In addition to the precipitates, constitutive elements were segregated near the dislocation lines, confirming the role of special boundaries as the preferential diffusion paths for precipitate growth. The EDS maps in **Supplementary Figure 4a₃** reveal the segregation and depletion of Fe, Mn, Cu, Ti, and Co at the grain boundaries surrounding the Ni_3Ti and Fe_2SiTi precipitates. This boundary segregation increases the energy barrier for dislocation motion, thus enhancing the overall material strength²². Fe diffused along the dislocation lines and accumulated in the interior regions, creating a region with a locally high Fe concentration, as indicated by the blue dashed line in **Supplementary Figures 4a** and **a₁**. Excess Fe in the grain can induce the transformation of the FCC phase into the body-centered cubic (BCC) phase by reducing the FCC phase stability²³. According to the diffraction pattern (**Supplementary Figure 4a₂**), the Fe-rich particles were identified as a new BCC structure formed during aging. The white arrow in **Supplementary Figure 4a₃** indicates that Cu was segregated at the grain boundaries as a distinct diffusion channel compared with Fe. Cu clusters were formed owing to the aforementioned Cu segregation, leading to Cu depletion at the grain-boundary regions (**Supplementary Figure 3a₁**).

(v) Compared with other constitutive elements, Cu has distinct characteristics because of its segregation at both precipitates and grain boundaries. Instead, as shown in the Cu EDS map in **Supplementary Figure 3a₁**, Cu formed isolated clusters approximately 10–20 nm in size located in the grain interior and in the vicinity of the precipitates. The chemical composition of the Cu clusters was analyzed via line scanning, as shown in **Supplementary Figures 6a–a₂**. The Cu cluster was enriched with approximately 32% Cu, whereas the Fe content was reduced to 32%. This preferential segregation of Cu is attributed to the

immiscibility of the Cu–Fe binary system at intermediate temperatures caused by the high positive mixing enthalpy between the two elements.

(vi) Independent of the Cu cluster, STEM and EDS images indicated a Cu–rich FCC precipitate (see Supplementary Figures 7a and a₁)—confirmed via bright–field imaging and nanobeam diffraction along the [011] axis, as shown in Supplementary Figures 7b and b₁. The SAED pattern (Supplementary Figure 7b₃) obtained from Supplementary Figure 7b reveals the superposition of the FCC pattern from the Cu–rich FCC and the matrix and the B2 pattern, which exhibits the Kurdjumov–Sachs orientation relationship. The HRTEM and FFT images in Supplementary Figures 7c–c₂ provide further insight into the selected areas. The B2 spots existed in a small area next to the Cu–rich FCC. However, no elemental segregation was observed apart from that of the Cu–rich FCC in Supplementary Figure 7a₁.

Comment #4:

3. Strengthening Mechanism from Spinodal Decomposition:

The authors have attributed the major contribution to strengthening to spinodal decomposition. To support this claim, it is crucial to provide evidence of the interaction between dislocations and the decomposed nano-particles from the TEM observation.

Response #4:

We would like to thank the reviewer for this comment. To reveal the interaction between the dislocation and the spinodal decomposed nanoparticles, the combined analyses of STEM bright–field imaging and EELS spectrum mapping were performed using a 2% strained sample (Figure 6). The wavy and serrated dislocations were observed in Figure 6a. Synchronized analyses of STEM dark–field image and EELS spectrum maps demonstrate that the dislocation pinning of the nano precipitates makes the wavy and serrated features (Figure 6a). We have included these findings in the main text. Please see page 16 lines 219 to 295:

Figure 6 shows that dislocation movement is hindered by the periodically spinodal decomposed nanoparticles, resulting in wavy and serrated dislocations at 2% local true strain. **Figure 6a** presents an overview of the deformed STEM bright-field image. **Figures 6b** and **b₁** demonstrate that the dislocation is impeded by the spinodal decomposed nanostructure.

Figure 6. The interaction of dislocation and nanoprecipitate under 2% strain. a STEM bright-field image reveals the wavy and serrated dislocations. **b** STEM dark-field image and **b₁** EELS spectrum maps of Ti-L_{2,3}, Ni-L_{2,3}, and Cu-L_{2,3}.

Comment #5:

The presence of twin structures after deformation raises questions regarding the potential contribution of twinning-induced plasticity to strengthening. The authors should provide a clear explanation of why they believe the strengthening mechanism is not primarily driven by twinning and offer supporting data or reasoning.

Response #5:

We are grateful for this constructive comment. We have conducted a deformation microstructure analysis to clarify the deformation behavior of the alloy. At 5% local true strain, nano twins were observed in TEM diffraction, and as deformation progressed to 15% and 25% local true strain, twin boundaries were observed in EBSD analysis. The fraction of twin boundaries increased from 7.4% at 15% ϵ_{tr} to 21.0% at 25% ϵ_{tr} , indicating that twinning acts as a strengthening mechanism during material deformation. We have addressed twinning-induced plasticity behavior of our alloy by including the additional discussion and Figure (in **Supplementary Figure 11**) in the revised manuscript. Please see page 13 lines 244 to 253:

2.3. Strengthening mechanism

A TEM analysis was conducted on a post-deformation sample at a local true strain (ϵ_{tr}) of 5% to investigate the exceptional combination of high strength and ductility in the aged sample. **Figures 5b–b₃** show bright- and dark-field images and the corresponding selected-area electron diffraction (SAED) patterns of the deformed aged sample. Profuse deformation nano-twins with an average thickness of 17.39 ± 10.98 nm (calculated from **Figure 5b**), which contributed to the increased strength through dynamic Hall-Petch strengthening^{34,35}, were detected in the early deformation stage. **EBSD investigation with twin boundary indexing on image quality images from local true strain levels (ϵ_{tr}) of 15% and 25% are also shown in **Supplementary Figure 11**. The twin boundary increased from 7.4% at the 15% ϵ_{tr} to 21.0% at the 25% ϵ_{tr} . As the ϵ_{tr} increases, deformation twins, initially confined to the non-recrystallized region, appear in the recrystallized region. The small twin spacing reduces the potential for dislocation pile-up, requiring more external stress to surpass and propagate across the twin boundary, thus contributing to work hardening^{36,67}. Therefore, the increase in twin boundary**

fraction indicates that twinning–induced plasticity is one of the main deformation mechanisms of the corresponding Fe–MEA. Phase transformation has not been detected upon the deformation due to the high stacking fault energy of the present FCC structure³⁸.

Comment #6:

Lastly, clarify the methodology behind plotting Fig. 3c, particularly considering that the TEM sample is from a 5% locally true strained sample. A detailed description of how this figure was generated will aid in the reader's understanding.

Addressing these points comprehensively and providing the necessary evidence and explanations will significantly enhance the paper's quality and its potential for acceptance in the publication.

Response #6:

Thank you for the comment. We have provided further clarification on the methodology for each result. **Figure 5b–b₂** displays deformation images at a 5% local true strain analyzed by TEM. **Figure 5c** illustrates the GND values plotted from EBSD data obtained with local true strain levels ranging from 5% to 25%.

The local true strain levels within the material were determined based on the deformation positions obtained from DIC data acquired through tensile testing. We have specified the analysis methods for each dataset in the manuscript and provided detailed descriptions in the methods section. Please see page 14 lines 257 to 259, page 19 lines 345 to 353, and page 20 lines 363 to 365:

In addition, in the presence of a heterogeneous structure resulting from a partially recrystallized area, deformation incompatibility between the recrystallized and non–recrystallized regions generates GNDs and subsequently hetero–deformation–induced (HDI) strengthening as an additional strengthening mechanism. As shown in **Figure 5c**, GND value development in the

non-recrystallized and recrystallized regions as a strain was obtained from EBSD results of the aged sample with the ϵ_{tr} of 5–25%; the accumulation of GNDs occurred predominantly in the recrystallized grains, leading to long-range internal stress, i.e., back stress³⁹.

Method

Microstructure characterization

The initial and deformed microstructures were characterized via EBSD. The initial, magnified initial, and deformed microstructure were obtained through EBSD measurement using a step size of 1.0 μm , 0.15 μm , and 0.2 μm , respectively. The results were analyzed using orientation imaging microscopy collection software (TSL OIM Analysis 7). To quantify the relative change in GND density from EBSD measurement, strain gradient theory^{58,59} was applied as shown in Equation (1):

$$\rho_{\text{GND}} = \frac{2\theta}{ub}, \quad (1)$$

where u is the step size (0.2 μm), b is the magnitude of Burgers vector, and θ is the average local misorientation angle measure in KAM maps.

For the analysis of the deformed microstructure, local true strain levels were obtained from DIC data (See Supplementary Figure 13).

Supplementary Figure 13. DIC image of local true strain distribution. The distribution of local true strain in the aged sample following fracture, as captured by the DIC technique.

Once again, we hope that the revised version can be further reviewed for publication in Nature Communications.

Best regards,
Hyoung Seop Kim
on behalf of all authors

REVIEWERS' COMMENTS

Reviewer #1 (Remarks to the Author):

All reviewer's comments and concerns have been addressed sufficiently, and the manuscripts' quality has been improved such that it can be considered for publication

Reviewer #2 (Remarks to the Author):

The authors have thoughtfully addressed the reviewers' comments. The manuscript can be accepted in its current form.

AUTHORS' POINT-BY-POINT RESPONSE TO THE REVIEWERS' COMMENTS
Nature Communications

Ref. No.: NCOMMS-23-63788-T

Dear Reviewers,

Thank you very much for your assessment and constructive comments on our manuscript (NCOMMS-23-63788-T) entitled *Periodic spinodal decomposition in double-strengthened medium-entropy alloy*. We realized that our previous manuscript had some unclear points. We have addressed these issues in the revised manuscript. In the revised manuscript, the responses to the reviewers are indicated in **blue**, and the revised parts were highlighted in **yellow**.

We hope that our revisions are satisfactory and clear.

With best regards on behalf of the author team,

Hyung Seop Kim

Response to reviewer #1:

Key results

The submitted manuscript suggests using matrix hardening through spinodal decomposition, inferring a level of strength gain with no impediment to toughness, as opposed to mere precipitation strengthening. Authors confidently use a large complimentary set of experimental techniques to address this research question, based on theoretical assumption made by other teams.

Comment #1:

Validity

The presented work does a good job of disseminating collected datasets. The measured elongation increase of ~28% shows that - to some extent - the proposed retention/increase of mechanical properties can be achieved.

However, some comments remain:

- Fig. 1 and associated text do not show comparable datasets for annealed and aged state, which makes it more difficult for the reader to follow argumentation about proposed mechanisms.

Response #1:

We are grateful for this constructive comment. We have added the EBSD image of the aged sample in **Supplementary Figure 2** to elucidate the microstructural evolution upon the aging process and improve the manuscript by providing further explanation. Please see page 4 lines 97 to 100:

Figures 1a and **b** show the electron backscatter diffraction (EBSD) phase and inverse pole figure (IPF) maps of the annealed sample. The phase of this sample was nearly fully FCC, with a small amount of martensite thermally induced by water quenching. Additionally, it

exhibited a partially recrystallized microstructure with fine grains and an average size of ~ 4.35 μm . The volume fraction of the recrystallized grains was approximately 30%, with predominant nucleation along the shear bands²¹ and grain boundaries²². The microstructure of the aged samples is shown in **Supplementary Figure 2**. Notably, it reveals that the annealed and aged samples have similar microstructures on the microscale because the aging temperature of 550 °C is not high enough to induce recrystallization.

Supplementary Figure 2. Aged sample. EBSD a phase and b IFP maps of the aged sample.

Comment #2:

Ext. figure 6b, showing the crystallographic information for area imaged in 6a, is hard to interpret due to its high contrast and poor image quality; a higher resolution broad beam SAED pattern of representative region could be helpful.

Response #2:

Thank you for your comment on Extended Fig. 6b. We aimed to capture the wavelength of the spinodal decomposition by analyzing the chemical composition along the $[001]_{\text{FCC}}$ direction, which exhibits a strong preferential direction for spinodal decomposition. For this purpose, we utilized the image in Extended Figure 6b, derived from atom probe tomography (APT) data, with the orientation information reprocessed using field ion microscopy. Regrettably, the corresponding image is poor compared to the electron backscatter diffraction (EBSD) pattern due to its origin from extracted ions in the APT data. In response to your suggestion, we have redrawn the image for better clarity, as shown in **Figure 3b**. This figure illustrates the hit detector map of the region delineated in **Figure 3a**, elucidating the crystallographic poles. The protocol described in the previous literature¹ was employed to obtain this image. We trust that this improved representation aids in your interpretation of the image and understanding of our purpose.

Figure 3. APT reconstruction of the aged sample. a APT reconstructed maps of alloying species in the studied alloy, showing the modulated structure. **b** Hit detector map of the region

marked by a black box, oriented along the [001] direction. Crystallographic main poles are indexed on the map. **c** One-dimensional concentration profile along the [001]_{FCC} and normalized composition profiles for Fe and Cu, with the maximum value of each element set to 1 for comparison in **c**.

1. Breen, A., Day, A., Lim, B., Davids, W. & Ringer, S. Revealing latent pole and zone line information in atom probe detector maps using crystallographically correlated metrics. *Ultramicroscopy* **243**, 113640 (2023).

Comment #3:

- A more in-depth explanation of the origins of the purported core-shell structure seen in the APT images is advised

Response #3:

We thank the reviewer for the feedback. The formation process of the core-shell structure unfolds as follows: During aging, the negative enthalpy of mixing between Ni and Ti drives the formation of a Ni-Ti-rich region. The composition of Ni is more than four times higher than that of Ti in this alloy. Most Ti combines with Ni to form the Ni-Ti-rich region, occupying all available vacancies. Consequently, the excess Ni, which tends to interact with Ti to form a unified structure with Ni-Ti bonding, aggregates around the Ni-Ti bonding to constitute the shell. However, the formation kinetics of the core-shell microstructure and nanoprecipitates in detail are beyond the scope of the current study. Please see page 7 lines 142 to 150:

“Based on the reconstructed APT interpretation, the isosurface maps for the two elements are shown in **Figure 2a-c**, with Ti and Ni strongly tangled with each other. Upon closer inspection, a core-shell structure was observed, with Ni covering Ti, as shown in **Figure 2a2**. The formation of the core-shell structure is influenced by the mixing enthalpy of the

constituent elements. During aging, the negative enthalpy of mixing between Ni and Ti drives the formation of a Ni–Ti–rich region. The atomic fraction of Ni is more than 4 times higher than that of Ti in this alloy. Most of the Ti combines with Ni to form the Ni–Ti–rich region, occupying all available vacancies. Consequently, the excess Ni, which tends to interact with Ti to form a uniform structure, aggregates around the Ni–Ti bonding to constitute the shell. The core–shell structure likely resulted from uphill diffusion, thus providing indirect evidence for spinodal decomposition²⁸. Though, the formation kinetics of the Ni–Ti core–shell nanoprecipitates in detail are beyond the scope of the current research.”

Comment #4:

- The insert image in Fig.2e is rather small, the axes text should be increased for readability; further the axis notation varies from at-pct to a.u., though it should be the same unit?

Response #4:

We appreciate the feedback regarding the insert image in Figure 2e. In response to your suggestions, we have taken measures to enhance the readability of the figure. Specifically, we have increased the font size of the axis text to ensure that it is easily legible.

To enhance the illustration of chemical composition fluctuation, particularly between Cu and Fe within the alloy system, we have added a separate subplot within Figure 3c that specifically emphasizes the fluctuation of Cu and Fe compositions. This addition aims to provide a more precise visualization of their distinct sinusoidal pattern. To facilitate a direct comparison between these compositions, we normalized their maximum values to 1. Therefore, the Y-axis is normalized intensity, arbitrary units (a.u.) are noted to represent these normalized values. Furthermore, we have enriched the figure legend with additional explanations concerning the normalization as follows:

Figure 3. APT reconstruction of the aged sample. **a** APT reconstructed maps of alloying species in the studied alloy, showing the modulated structure. **b** Hit detector map of the region marked by black box, oriented along the [001] direction. Crystallographic main poles are indexed on the map. **c** One-dimensional concentration profile along the [001]_{FCC} and Normalized composition profiles for Fe and Cu, with the maximum value of each element set to 1 for comparison in **c**.

Comment #5:

Data and methodology

Generally, significant efforts were undertaken in support of this work, involving a wide range of characterization techniques, as well as mechanical testing.

- Induction melting in graphite mold involves the potential for significant carbon pickup, which should be a point of consideration in a Fe-based alloy system

Response #5:

As the reviewer has concerned, the carbon concentration in Fe-based alloys is an important factor in microstructure and mechanical properties, including the stability of the FCC matrix and the formation of precipitate phases. We have conducted carbon/sulfur determination using (LECO/CS844 C/S Analyzer, Leco Corporation, St. Joseph, USA) to analyze carbon content. In our Fe-based alloy, the carbon concentration of 107 ± 5.6 ppm is closely aligned with the levels found in extremely low carbon steels². This specific concentration is below the threshold for influencing the formation of precipitating phases and carbides or changing phase stability that could significantly alter material properties. This minimal carbon level ensures that the primary effects of carbon, such as phase stabilization and minor adjustments to mechanical properties, do not lead to the formation of carbides or other precipitates that could detrimentally affect the alloy's performance.

2. Koga, N., Kanehira, Y., Huyen, P.T.T., Hori, K. & Umezawa, O. Effect of aging on low-temperature tensile properties of ultra-low carbon steel. *ISIJ International* **61**, 2308-2316 (2021).

Comment #6:

- How were GND measurements conducted?

Response #6:

Thank you for the comment. We calculated the GND from the KAM values obtained from EBSD data (presented in **Figure 5c**). We provide further details on how we measured the GND value in the Method section. Please see page 19 lines 345 to 353:

The annealed, aged, magnified annealed, and deformed microstructure were obtained through EBSD measurement using a step size of 1.0 μm , 1.0 μm , 0.15 μm , and 0.2 μm , respectively. The results were analyzed using orientation imaging microscopy collection software (TSL OIM Analysis 7). To quantify the relative change in GND density from EBSD measurement, strain gradient theory^{3,4} was applied as shown in Equation (1):

$$\rho_{\text{GND}} = \frac{2\theta}{ub}, \quad (1)$$

where u is the step size (0.2 μm), b is the magnitude of the Burgers vector, and θ is the average local misorientation angle measured in KAM maps.

3. Gao, H., Huang, Y., Nix, W.D. & Hutchinson, J.W. Mechanism-based strain gradient plasticity - I. Theory. *J Mech Phys Solids* **47**, 1239-1263 (1999).
4. Kubin, L.P. & Mortensen, A. Geometrically necessary dislocations and strain-gradient plasticity: a few critical issues. *Scripta Materialia* **48**, 119-125 (2003).

Comment #7:

- How many mechanical tests were conducted, including replicates? This is crucial to know for assessing statistical significance of results.

Response #7:

We thank the reviewer for this constructive comment. To ensure reproducibility, we performed the tensile test at least three times for each condition. The stress-strain curves of the reproducibility test are presented in Figure R1. While minor variations were observed in the reproducibility experiments, they remained within acceptable limits. The significant improvement in strength observed between the annealed and aged samples in every condition cannot solely be attributed to precipitation; it is also influenced by the spinodal decomposed structure described in our manuscript. We have added a note about reproducibility in the Method section. Please see page 19 lines 341 to 342:

Mechanical-property tests

A flat dog-bone-shaped sample with a thickness of 1.5 mm, width of 2.5 mm, and gauge length of 5.0 mm was cut along the rolling direction. A room-temperature tensile test utilizing digital image correlation (ARAMIS M12, GOM Optical Measuring Techniques, Germany) was conducted using a universal testing machine (Instron 1361, Instron Corp., USA) at a strain rate of $1 \times 10^{-3} \text{ s}^{-1}$. To ensure reproducibility, tensile tests were conducted at least three times for each condition.

Figure R1. **a** engineering stress–strain curve of annealed samples and **b** engineering stress–strain curve of aged samples.

Comment #8:

- A further question is the originality aspect of the alloy design; there is little mention of how the authors arrived at the proposed composition, whether this was informed by statistical and computational modeling.

Related, the employed thermal annealing processing steps are not referenced by prior work or open literature, specifically the aging temperature leading to successful spinodal decomposition.

Response #8:

We appreciate the reviewer's insightful inquiries concerning the originality of our alloy design and the specifics of the heat treatment process. Addressing the first point on alloy composition, our selection of the complex elemental alloy system is grounded in thermodynamic principles. The thermodynamic principle, $\Delta G = \Delta H - T\Delta S$, where ΔG represents Gibbs free energy, ΔH is the mixing enthalpy, ΔS is the mixing entropy, and T denotes temperature. The introduction of a large number of constituent elements, while beneficial for increasing the entropy, also has the potential to increase the mixing enthalpy⁵. The current Fe_{61.75}Ni_{14.25}Co_{7.6}Mn_{7.6}Ti_{2.85}Si_{0.95}Cu_{4.5}Al_{0.5} MEA alloy has been designed as a branch of the Fe–Ti–Si–based nanoprecipitation–strengthened medium–entropy alloys inspired by the previous alloys Fe₆₅Ni₁₅Mn₈Co₈Ti₃Si₁⁶ and Fe₆₈Ni₁₀Mn₁₀Co₁₀Ti_{1.5}Si_{0.5}⁷. An extraordinarily high mechanical strength has been reported after the formation of Fe₂SiTi and Ni₃Ti nanoprecipitates, yet high ductility as the result of the presence of non–shearable precipitates^{6–14}. On the other hand, the strong ordering tendencies of Al, along with most of the other alloying elements, make the ordering transition of primary importance. A sufficient concentration of Al encouraged a series of phase transformations, including the formation of coherent mixtures of B2 and partially ordered aluminides in a body–centered cubic (BCC) solid–solution matrix^{15–17}. In the presence of Al–containing B2 or BCC phase, the mechanical strength incredibly increases at the expense of ductility. Therefore, to avoid these phase transformations leading to ductility reduction, yet using the advantage of lattice expansion and solid solution strengthening, a low concentration of 0.5 at% was chosen for the Al element. In the Fe–based alloy and CrMnFeCoNi alloy systems, previous literature has revealed microstructural change with the addition of Cu^{18–21}. The addition of Cu above 5% can induce Cu–rich precipitation or microscale phase separation²¹. Therefore, in our study, we have strategically limited the Cu addition to below 5% to induce the desired nanostructures without promoting excessive phase separation.

By controlling the composition and microstructure through thermal processing associated with the phase transformations, this system provides a wide range of potential compounds and

associated microstructures. Thus, the main concept of the present work is to design a stable FCC matrix to enhance its sensitivity to spinodal decomposition. Based on the thermodynamic calculations and the equilibrium phase diagram presented in **Supplementary Figure 1**, the presence of Fe₂SiTi, Ni₃Ti, B2, and Cu-rich FCC phases is possible upon a suitable heat treatment in the range of ~450–600 °C. Therefore, by applying the similar aging treatment we have done in our previous work (at 550 °C for 5 hours⁶, as shown in the differential scanning calorimetry (DSC) analysis in **Supplementary Figure 14**), all the aforementioned precipitates have been formed in our microstructure. This results in an excellent combination of UTS and uniform elongation compared to all these reported alloys. Besides, the spectacular characteristic of the core-shell microstructure classifies this Fe-MEA as an alloy with high potential for high-strength applications.

Therefore, such a complex multi-heterogeneous microstructure of the current Fe-MEA is apparently distinct from the simpler microstructure of the previous one⁶. The current manuscript also intensifies the role of alloying elements on the phase stability of the initial microstructure, showing how the microstructure converts from a solid solution state to a precipitate/spinodal decomposed structure strengthened one with only a small change in compositional elements compared to the previous Fe₆₅Ni₁₅Mn₈Co₈Ti₃Si₁ MEA⁶.

We have revised the **introduction** based on this information and added it as **Supplementary Note 1. Alloy Design Strategy**, as follows:

Introduction

To realize spinodal decomposition in multi-principal element alloys, we designed a new ferrous medium-entropy alloy (Fe_{61.75}Ni_{14.25}Co_{7.6}Mn_{7.6}Ti_{2.85}Si_{0.95}Cu_{4.5}Al_{0.5} denoted as Fe-MEA) by adding the minor elements Cu and Al. This deliberate compositional modification introduces the formation of a miscibility gap and subsequent spinodal decomposition. **This complex compositional strategy not only increases the entropy of the alloy but also enhances its enthalpy, thereby improving the probability of spinodal decomposition¹⁸. Cu, which has a large miscible gap with Fe—the principal component of the alloy—was added 4.5%. Adding over 5% of Cu is avoided due to the formation of microscale phase separation and immoderate Cu-rich precipitates¹⁹. Al was deliberately selected at a concentration of 0.5 at% to strategically leverage the benefits of lattice expansion and solid solution strengthening. Both Cu and Al are**

widely known to increase the stacking fault energy, preventing the generation of transformation-induced plasticity²⁰. Based on the phase diagram (Supplementary Figure 1), the presence of Fe₂SiTi, Ni₃Ti, B2, and Cu-rich FCC phases is possible upon a suitable heat treatment in the range of ~450–600 °C; thus the appropriate heat treatment should be performed to achieve the desired spinodal decomposition and the formation of multiple dynamic precipitates simultaneously (detail describes in Supplementary Note 1 and method section). Consequently, Fe–MEA utilizing spinodal decomposition shows a doubled strength without notable ductility loss. Leveraging spinodal decomposition holds significant promise for advancing alloy design strategies and overcoming the long-standing challenge of balancing strength and ductility in H/MEAs, opening up new horizons for developing next-generation metallic materials.

Supplementary Note 1. Alloy design strategy

The design of this alloy strategically exploits the diversity of constituent elements and their enthalpy contributions to promote spinodal decomposition. The thermodynamic principle, $\Delta G = \Delta H - T\Delta S$, where ΔG represents Gibbs free energy, ΔH is the mixing enthalpy, ΔS is the mixing entropy, and T denotes temperature, describes the significance of the high entropy value in M/HEAs. This high entropy stabilizes the Gibbs free energy, promoting a solid solution state in M/HEAs. Contrary to the beneficial "high entropy effect," an increased number of constituent elements can elevate the enthalpy, thereby enhancing the likelihood of spinodal decomposition².

The current Fe_{61.75}Ni_{14.25}Co_{7.6}Mn_{7.6}Ti_{2.85}Si_{0.95}Cu_{4.5}Al_{0.5} MEA alloy has been designed as a branch of Fe-Ti-Si-based nanoprecipitation-strengthened medium-entropy alloys inspired by the previous alloy Fe₆₅Ni₁₅Mn₈Co₈Ti₃Si₁³ and Fe₆₈Ni₁₀Mn₁₀Co₁₀Ti_{1.5}Si_{0.5}⁴. An extraordinarily high mechanical strength has been reported after the formation of Fe₂SiTi and Ni₃Ti nanoprecipitates, yet high ductility resulting from the presence of non-shearable precipitates³⁻¹¹. On the other hand, the strong ordering tendencies of Al, along with most of the other alloying elements, make the ordering transition of primary importance. A sufficient concentration of Al encouraged a series of phase transformations, including the formation of coherent mixtures of B2 and partially ordered aluminides in a body-centered cubic (BCC)

solid–solution matrix¹²⁻¹⁴. In the presence of Al–containing B2 or BCC phase, the mechanical strength incredibly increases at the expense of ductility. Therefore, to avoid these phase transformations leading to ductility reduction, while using the advantage of lattice expansion and solid solution strengthening, a low concentration of 0.5% was chosen for the Al element. The Fe–Cu binary system is widely known for its positive mixing enthalpy, which significantly tends towards phase separation. In the Fe–based alloy and CrMnFeCoNi alloy systems, previous literature has revealed microstructural changes with the addition of Cu¹⁵⁻¹⁸. The addition of Cu above 5 at% can induce Cu–rich precipitation or microscale phase separation¹⁸. Therefore, in our study, we have strategically limited the Cu addition to below 5% to induce the desired nanostructures without promoting excessive phase separation.

By controlling the composition and microstructure through thermal processing associated with the phase transformations, this system provides a wide range of potential compounds and associated microstructures. Thus, the main concept of the present work is to design a stable FCC matrix to enhance its sensitivity to spinodal decomposition. Based on the thermodynamic calculations and the equilibrium phase diagram presented in Supplementary Figure 1, the presence of Fe₂SiTi, Ni₃Ti, B2, and Cu–rich FCC phases is possible upon a suitable heat treatment in the range of ~450–600 °C. Therefore, by applying a similar aging treatment as we have done in our previous work³ (at 550 °C for 5 hours), as shown in the differential scanning calorimetry (DSC) analysis in Supplementary Figure 14, all the aforementioned precipitates have been formed in our microstructure.

5. Luan, H.W. *et al.* Spinodal decomposition and the pseudo-binary decomposition in high-entropy alloys. *Acta Mater.* **248** (2023).
6. Fillon, A. *et al.* Influence of severe plastic deformation on the precipitation hardening of a FeSiTi steel. *J. Mater. Sci.* **47**, 7939-7945 (2012).
7. Perrier, M., Bouaziz, O., Brechet, Y., Deschamps, A. & Donnadieu, P. Mechanical properties of low carbon steel hardened by the Fe₂SiTi phase at high volume fraction. *J. Phys.: Conf. Ser.* **240**, 012095 (2010).
8. Knowles, A.J. *et al.* Development of Ni-free Mn-stabilised maraging steels using Fe₂SiTi precipitates. *Acta Mater.* **174**, 260-270 (2019).

9. Perrier, M. *et al.* Precipitation sequence and kinetics in an Fe-Si-Ti alloy. *Solid State Phenom.* **172**, 833-838 (2011).
10. Perrier, M. *et al.* Characterization and modeling of precipitation kinetics in a Fe-Si-Ti alloy. *Metall. Mater. Trans. A* **43**, 4999-5008 (2012).
11. Löffler, F., Palm, M. & Sauthoff, G. Iron-Rich Iron-Titanium-Silicon Alloys with Strengthening Intermetallic Laves Phase Precipitates. *Steel Res. Int.* **75**, 766-772 (2004).
12. Jack, D. & Honeycombe, R. Age hardening of an Fe-Ti-Si alloy. *Acta Metall.* **20**, 787-796 (1972).
13. Haftlang, F., Seol, J.B., Zargaran, A., Moon, J. & Kim, H.S. Chemical core-shell metastability-induced large ductility in medium-entropy maraging and reversion alloys. *Acta Mater.* **256**, 119115 (2023).
14. Haftlang, F. *et al.* Simultaneous effects of deformation-induced plasticity and precipitation hardening in metastable non-equiatomic FeNiCoMnTiSi ferrous medium-entropy alloy at room and liquid nitrogen temperatures. *Scr. Mater.* **202** (2021).
15. Santodonato, L.J. *et al.* Deviation from high-entropy configurations in the atomic distributions of a multi-principal-element alloy. *Nat. commun* **6**, 5964 (2015).
16. Singh, S., Wanderka, N., Murty, B., Glatzel, U. & Banhart, J. Decomposition in multi-component AlCoCrCuFeNi high-entropy alloy. *Acta Mater.* **59**, 182-190 (2011).
17. Santodonato, L.J., Liaw, P.K., Unocic, R.R., Bei, H. & Morris, J.R. Predictive multiphase evolution in Al-containing high-entropy alloys. *Nat. commun* **9**, 4520 (2018).
18. Ma, X. *et al.* Spinodal decomposition of precipitation hardening Fe-17Cr-4Ni-4Cu stainless steel at 475 C. *Materials and technology* **56**, 193–199-193–199 (2022).
19. Fiocchi, J., Mostaed, A., Coduri, M., Tuissi, A. & Casati, R. Development and characterization of a novel high entropy alloy strengthened through concurrent spinodal decomposition and precipitation. *J. Alloy. Compd.* **947**, 169706 (2023).
20. Zhang, Y. *et al.* Concurrence of spinodal decomposition and nano-phase precipitation in a multi-component AlCoCrCuFeNi high-entropy alloy. *J. Mater. Res. Technol* **8**, 726-736 (2019).

21. Du, C. *et al.* Effect of Cu on the strengthening and embrittling of an FeCoNiCr-xCu HEA. *Mater. Sci. Eng. A* **832**, 142413 (2022).

Response to reviewer #2:

Comments on "Periodic spinodal decomposition in double-strengthened medium-entropy alloy".

This paper presents significant findings on the remarkable strengthening effects observed in a ferrous medium-entropy alloy ($\text{Fe}_{61.75}\text{Ni}_{14.25}\text{Co}_{7.6}\text{Mn}_{7.6}\text{Ti}_{2.85}\text{Si}_{0.95}\text{Cu}_{4.5}\text{Al}_{0.5}$), with notable attention given to the achievement of a high tensile stress value. The feedback provided by the reviewer has been presented, and addressing these points properly will contribute to the paper's overall quality and its potential acceptance for publication.

Comment #1:

1. GND Map in Fig. 1b:

While conventional EBSD measurements have been employed to explore the GND density, the reviewer raises concerns about its accuracy. To improve the precision of GND density determination, it is recommended to conduct a crosscourt analysis. It is preferably utilizing the KAM profile here.

Response #1:

We sincerely appreciate the reviewer's valuable feedback. Our intention in Figure 2b is to illustrate the difference in dislocation density between recrystallized and non-recrystallized regions. The reviewer's concern regarding the accuracy of presenting GND density measured from EBSD in a single image has been noted. Therefore, we have decided to replace the GND map with a KAM map in Figure 2b, thereby ensuring a more precise representation of the data. Additionally, we have made corresponding edits to the manuscript. Please see page 4 lines 96 to 97:

Figures 1a and **b** show the electron backscatter diffraction (EBSD) phase and inverse pole figure (IPF) maps of the annealed sample. The phase of this sample was nearly fully FCC,

with a small amount of martensite thermally induced by water quenching. Additionally, it exhibited a partially recrystallized microstructure with fine grains and an average size of $\sim 4.35 \mu\text{m}$. The volume fraction of the recrystallized grains was approximately 30%, with predominant nucleation along the shear bands²² and grain boundaries²³. The magnified images of the annealed sample in **Figures 1b₁** and **b₂** reveal a discernible difference in **kernel average misorientation (KAM) value** between the recrystallized and non-recrystallized regions.

Figure 1. Compositional heterogeneities of the Fe–MEA. EBSD **a** phase and **b** IPF maps of the aged samples. Magnified **b₁** IPF and **b₂** KAM maps of an aged sample. **c** TEM bright–field image of the initial microstructure of the aged sample, along with STEM images and corresponding EDS maps of the **c₁** recrystallized region and **c₂** non–recrystallized region magnified in **c**.

Comment #2:

2. Analysis of the Moire Pattern in Fig. 2d:

The Moire pattern analysis in Fig. 2d has been questioned due to the observed extension of the pattern beyond the expected particle size of approximately 5nm. This suggests the possibility of Moire pattern generation from the overlap of two grains within the TEM sample, potentially due to excessive sample thickness. To address this concern, a discussion on the influence of sample thickness on Moire pattern observations is necessary.

Response #2:

We appreciate the reviewer’s critical comment. Even though the formation of nano–sized precipitates is illustrated in **Figures 1c₁** and **c₂**, the specimen thickness effect on the obtained phase contrast image hinders the direct understanding of the precipitate formation behavior. We have performed additional TEM observations to clarify the specimen thickness–dependent Moiré pattern generation, as shown in **Supplementary Figure 10**. Two specimen thicknesses (44 nm and 54 nm) were selected for comparison. The specimen thickness was measured using the low–loss spectra in electron energy loss spectroscopy (EELS). The detailed method of measuring specimen thickness is given in the previous report²². The STEM bright–field image in **Supplementary Figure 10b**, where the thickness is about 44 nm, shows isolated and overlapped distributions of the nanoprecipitates. This tendency coincides with the EDS Ti map in **Supplementary Figure 10b₁**. The HRTEM image was obtained in the same area (**Supplementary Figure 10b₂**). The Moiré patterns caused by the overlap of B2 and matrix

reflect the projected distribution of B2 particles in the matrix: some particles are isolated, and others are overlapped. However, the nanoprecipitates mostly overlap in the thicker region (54 nm) (Supplementary Figures 10c and c₁). The Moiré patterns in this area reveal the overlapping of B2 particles in Supplementary Figures 10c₃. We have added this discussion to the revised manuscript.

Supplementary Figure 10. Specimen thickness–dependent Moiré patterns. a STEM bright–field image showing variation in specimen thickness; **a₁** EELS low–loss spectra corresponding the positions **b** and **c** in **a**. **b** high magnification image of the position **b** in **a**; **b₁** EDS maps of Ti and Cu; **b₂** HR image of the corresponding region. **c** high magnification image of the position **c** in **a**; **c₁** EDS maps of Ti and Cu; **c₂** HR image of the corresponding region.

22. Heo, Y.U. Comparative study on the specimen thickness measurement using EELS and CBED methods. *Appl Microsc* 50, 8 (2020).

Comment #3:

Additionally, the authors should clarify the schematic diagram presented in Fig. 2f. A detailed explanation of the intended information conveyed by the middle figure of Fig. 2f will enhance the reader's comprehension.

Response #3:

Thank you for the reviewer's comment about the schematic image in Figure 2f. Our aged samples exhibit a complex microstructure characterized not only by a spinodal decomposed structure but also by a variety of precipitate phases and compositional heterogeneities. These features undoubtedly contribute to precipitation hardening and influence the material's mechanical properties. However, they are not the primary focus of our investigation. Our paper centers on elucidating the role and mechanisms of spinodal hardening in enhancing the mechanical properties of our alloy system. Therefore, the additional precipitate phases and compositional heterogeneity in detail are discussed in the supplementary materials. The supplementary section is intended to offer interested readers further insights into the microstructural complexity of our aged samples, including a thorough analysis of the diverse precipitate types and their distribution, as well as the overall compositional variability. For a better understanding of our manuscript, we have added summarized explanations of precipitate phases and compositional heterogeneities in the manuscript. Please see page 5 lines 109 to 117:

The TEM/STEM and atom probe tomography (APT) images revealed several distinct compositional heterogeneities in the formation of precipitates with distinctive elemental distributions. This includes the discovery of Fe_2SiTi and Ni_3Ti nanoprecipitates, which are abundant precipitates in the alloy system, alongside the observation of $\text{Ni}_3(\text{Ti}, \text{Si})_2$, a metastable

phase prone to transformation under prolonged aging²⁴. Additionally, elemental segregation near dislocations and at grain boundaries was identified, highlighting the diffusion paths that favor precipitation growth. The formation of distinct phases and clusters, such as Fe-rich BCC, Cu-rich FCC precipitates, and Cu clusters—are explained in detail in Supplementary Figures 3–7 and Supplementary Note 2.

Supplementary Note 2. Multiple dynamic precipitates in the aged sample

To clarify the microstructural features of the aged sample, TEM and APT analyses were conducted to investigate the nanoscale microstructural evolution during aging. The STEM image and corresponding EDS maps revealed several distinct compositional heterogeneities in the form of precipitates with distinctive elemental distributions, as follows:

(i) The presence of a Fe₂SiTi precipitate with a hexagonal close-packed (HCP) structure (P63/mmc, $a = 4.77 \text{ \AA}$, $c = 7.74 \text{ \AA}$) in the Fe–Mn–Cu–depleted area was confirmed by the SAED pattern corresponding to the area indicated by the white dashed line in the STEM and TEM dark-field images (see Supplementary Figures 3a and a2). Additionally, the reconstructed APT concentration profile (Supplementary Figure 3b₁) of the Fe₂SiTi precipitate in (Supplementary Figure 3b) indicated high concentrations of Si and Ti across the precipitate.

(ii) In addition to Fe₂SiTi, as indicated by white arrows in the EDS map in Supplementary Figure 3a₁, the presence of Ni- and Ti-rich areas (indicated by red arrows in the Ni EDS map in Supplementary Figure 3a₁) suggested the formation of Ni₃Ti nanoprecipitates. A perspective illustration of the Ni₃Ti precipitate is shown by the green dashed line in the STEM and dark-field images in Supplementary Figures 4a and a₁; the Ni₃Ti precipitate is enriched in Ni and Ti and depleted in Fe, Cu, Mn, and Co, as confirmed by the corresponding EDS results in Supplementary Figure 4a₃. The FFT in Supplementary Figures 5a–a₂ identifies the Ni₃Ti precipitate as an η -D0₂₄ nanoprecipitate with an HCP structure (P63/mmc, $a = 5.074 \text{ \AA}$, and $c = 8.31 \text{ \AA}$) along the $[2\bar{1}\bar{1}0]_{\eta}$ zone axis, with the matrix identified as FCC along the $[011]$ zone axis. The Ni₃Ti precipitate exhibits semi-coherency with the FCC matrix in the Nishiyama–Wassermann orientation with $(1\bar{1}1)_{\text{FCC}} \parallel (0001)_{\eta}$ and

$[011]_{\text{FCC}} \parallel [2\bar{1}\bar{1}0]_{\eta}$ ¹⁹. Ni_3Ti and Fe_2SiTi precipitates were abundant in the alloys, typically with average sizes of 20–200 nm.

(iii) More interestingly, the $\text{Ni}_3(\text{Ti}, \text{Si})_2$ nanoprecipitate—indicated by the yellow arrow in the Ni map in **Supplementary Figure 3a₁**—was confirmed as HCP (P63/mmc, $a = 4.57 \text{ \AA}$, and $c = 7.96 \text{ \AA}$) along the $[01\bar{1}0]_{\text{HCP}}$ zone axis, as shown in **Supplementary Figures 5b** and **b₁**. $\text{Ni}_3(\text{Ti}, \text{Si})_2$ is a metastable phase commonly observed in Ni–Ti–based alloys^{20,21}. The metastable phase can coarsen and decompose at high aging temperatures with long aging times to form the stable phase Ni_3Ti ²¹.

(iv) In addition to the precipitates, constitutive elements were segregated near the dislocation lines, confirming the role of special boundaries as the preferential diffusion paths for precipitate growth. The EDS maps in **Supplementary Figure 4a₃** reveal the segregation and depletion of Fe, Mn, Cu, Ti, and Co at the grain boundaries surrounding the Ni_3Ti and Fe_2SiTi precipitates. This boundary segregation increases the energy barrier for dislocation motion, thus enhancing the overall material strength²². Fe diffused along the dislocation lines and accumulated in the interior regions, creating a region with a locally high Fe concentration, as indicated by the blue dashed line in **Supplementary Figures 4a** and **a₁**. Excess Fe in the grain can induce the transformation of the FCC phase into the body-centered cubic (BCC) phase by reducing the FCC phase stability²³. According to the diffraction pattern (**Supplementary Figure 4a₂**), the Fe-rich particles were identified as a new BCC structure formed during aging. The white arrow in **Supplementary Figure 4a₃** indicates that Cu was segregated at the grain boundaries as a distinct diffusion channel compared with Fe. Cu clusters were formed owing to the aforementioned Cu segregation, leading to Cu depletion at the grain-boundary regions (**Supplementary Figure 3a₁**).

(v) Compared with other constitutive elements, Cu has distinct characteristics because of its segregation at both precipitates and grain boundaries. Instead, as shown in the Cu EDS map in **Supplementary Figure 3a₁**, Cu formed isolated clusters approximately 10–20 nm in size located in the grain interior and in the vicinity of the precipitates. The chemical composition of the Cu clusters was analyzed via line scanning, as shown in **Supplementary Figures 6a–a₂**. The Cu cluster was enriched with approximately 32% Cu, whereas the Fe content was reduced to 32%. This preferential segregation of Cu is attributed to the

immiscibility of the Cu–Fe binary system at intermediate temperatures caused by the high positive mixing enthalpy between the two elements.

(vi) Independent of the Cu cluster, STEM and EDS images indicated a Cu–rich FCC precipitate (see Supplementary Figures 7a and a₁)—confirmed via bright–field imaging and nanobeam diffraction along the [011] axis, as shown in Supplementary Figures 7b and b₁. The SAED pattern (Supplementary Figure 7b₃) obtained from Supplementary Figure 7b reveals the superposition of the FCC pattern from the Cu–rich FCC and the matrix and the B2 pattern, which exhibits the Kurdjumov–Sachs orientation relationship. The HRTEM and FFT images in Supplementary Figures 7c–c₂ provide further insight into the selected areas. The B2 spots existed in a small area next to the Cu–rich FCC. However, no elemental segregation was observed apart from that of the Cu–rich FCC in Supplementary Figure 7a₁.

Comment #4:

3. Strengthening Mechanism from Spinodal Decomposition:

The authors have attributed the major contribution to strengthening to spinodal decomposition. To support this claim, it is crucial to provide evidence of the interaction between dislocations and the decomposed nano-particles from the TEM observation.

Response #4:

We would like to thank the reviewer for this comment. To reveal the interaction between the dislocation and the spinodal decomposed nanoparticles, the combined analyses of STEM bright–field imaging and EELS spectrum mapping were performed using a 2% strained sample (Figure 6). The wavy and serrated dislocations were observed in Figure 6a. Synchronized analyses of STEM dark–field image and EELS spectrum maps demonstrate that the dislocation pinning of the nano precipitates makes the wavy and serrated features (Figure 6a). We have included these findings in the main text. Please see page 16 lines 219 to 295:

Figure 6 shows that dislocation movement is hindered by the periodically spinodal decomposed nanoparticles, resulting in wavy and serrated dislocations at 2% local true strain. **Figure 6a** presents an overview of the deformed STEM bright-field image. **Figures 6b** and **b₁** demonstrate that the dislocation is impeded by the spinodal decomposed nanostructure.

Figure 6. The interaction of dislocation and nanoprecipitate under 2% strain. a STEM bright-field image reveals the wavy and serrated dislocations. **b** STEM dark-field image and **b₁** EELS spectrum maps of Ti-L_{2,3}, Ni-L_{2,3}, and Cu-L_{2,3}.

Comment #5:

The presence of twin structures after deformation raises questions regarding the potential contribution of twinning-induced plasticity to strengthening. The authors should provide a clear explanation of why they believe the strengthening mechanism is not primarily driven by twinning and offer supporting data or reasoning.

Response #5:

We are grateful for this constructive comment. We have conducted a deformation microstructure analysis to clarify the deformation behavior of the alloy. At 5% local true strain, nano twins were observed in TEM diffraction, and as deformation progressed to 15% and 25% local true strain, twin boundaries were observed in EBSD analysis. The fraction of twin boundaries increased from 7.4% at 15% ϵ_{tr} to 21.0% at 25% ϵ_{tr} , indicating that twinning acts as a strengthening mechanism during material deformation. We have addressed twinning-induced plasticity behavior of our alloy by including the additional discussion and Figure (in **Supplementary Figure 11**) in the revised manuscript. Please see page 13 lines 244 to 253:

2.3. Strengthening mechanism

A TEM analysis was conducted on a post-deformation sample at a local true strain (ϵ_{tr}) of 5% to investigate the exceptional combination of high strength and ductility in the aged sample. **Figures 5b–b₃** show bright- and dark-field images and the corresponding selected-area electron diffraction (SAED) patterns of the deformed aged sample. Profuse deformation nano-twins with an average thickness of 17.39 ± 10.98 nm (calculated from **Figure 5b**), which contributed to the increased strength through dynamic Hall-Petch strengthening^{34,35}, were detected in the early deformation stage. **EBSD investigation with twin boundary indexing on image quality images from local true strain levels (ϵ_{tr}) of 15% and 25% are also shown in **Supplementary Figure 11**. The twin boundary increased from 7.4% at the 15% ϵ_{tr} to 21.0% at the 25% ϵ_{tr} . As the ϵ_{tr} increases, deformation twins, initially confined to the non-recrystallized region, appear in the recrystallized region. The small twin spacing reduces the potential for dislocation pile-up, requiring more external stress to surpass and propagate across the twin boundary, thus contributing to work hardening^{36,67}. Therefore, the increase in twin boundary**

fraction indicates that twinning–induced plasticity is one of the main deformation mechanisms of the corresponding Fe–MEA. Phase transformation has not been detected upon the deformation due to the high stacking fault energy of the present FCC structure³⁸.

Comment #6:

Lastly, clarify the methodology behind plotting Fig. 3c, particularly considering that the TEM sample is from a 5% locally true strained sample. A detailed description of how this figure was generated will aid in the reader's understanding.

Addressing these points comprehensively and providing the necessary evidence and explanations will significantly enhance the paper's quality and its potential for acceptance in the publication.

Response #6:

Thank you for the comment. We have provided further clarification on the methodology for each result. **Figure 5b–b₂** displays deformation images at a 5% local true strain analyzed by TEM. **Figure 5c** illustrates the GND values plotted from EBSD data obtained with local true strain levels ranging from 5% to 25%.

The local true strain levels within the material were determined based on the deformation positions obtained from DIC data acquired through tensile testing. We have specified the analysis methods for each dataset in the manuscript and provided detailed descriptions in the methods section. Please see page 14 lines 257 to 259, page 19 lines 345 to 353, and page 20 lines 363 to 365:

In addition, in the presence of a heterogeneous structure resulting from a partially recrystallized area, deformation incompatibility between the recrystallized and non–recrystallized regions generates GNDs and subsequently hetero–deformation–induced (HDI) strengthening as an additional strengthening mechanism. As shown in **Figure 5c**, GND value development in the

non-recrystallized and recrystallized regions as a strain was obtained from EBSD results of the aged sample with the ϵ_{tr} of 5–25%; the accumulation of GNDs occurred predominantly in the recrystallized grains, leading to long-range internal stress, i.e., back stress³⁹.

Method

Microstructure characterization

The initial and deformed microstructures were characterized via EBSD. The initial, magnified initial, and deformed microstructure were obtained through EBSD measurement using a step size of 1.0 μm , 0.15 μm , and 0.2 μm , respectively. The results were analyzed using orientation imaging microscopy collection software (TSL OIM Analysis 7). To quantify the relative change in GND density from EBSD measurement, strain gradient theory^{58,59} was applied as shown in Equation (1):

$$\rho_{\text{GND}} = \frac{2\theta}{ub}, \quad (1)$$

where u is the step size (0.2 μm), b is the magnitude of Burgers vector, and θ is the average local misorientation angle measure in KAM maps.

For the analysis of the deformed microstructure, local true strain levels were obtained from DIC data (See Supplementary Figure 13).

Supplementary Figure 13. DIC image of local true strain distribution. The distribution of local true strain in the aged sample following fracture, as captured by the DIC technique.

Once again, we hope that the revised version can be further reviewed for publication in Nature Communications.

Best regards,
Hyoung Seop Kim
on behalf of all authors